# Measurement Report: Optical Characterization, Seasonality, and Sources of Brown Carbon in Fine Aerosols from Tianjin, North China: Year-round Observations

**Zhichao Dong[1], Chandra Mouli Pavuluri[1]\*, Peisen Li[1], Zhanjie Xu[1], Junjun Deng[1], Xueyan Zhao[1], Xiaomai Zhao[1], Pingqing Fu[1], Cong-Qiang Liu[1]**

[1]Institute of Surface-Earth System Science, School of Earth System Science, Tianjin University, Tianjin 300072, China

*Correspondence to:* Chandra Mouli Pavuluri (cmpavuluri@tju.edu.cn)

## Abstract

To investigate the optical characteristics and sources of brown carbon (BrC) in North China, where the atmospheric aerosol loadings are high and have severe impacts on the Earth's climate system, we collected fine aerosols ($PM_{2.5}$) at an urban site in Tianjin over a 1-year period. We measured the ultraviolet (UV) light absorption and excitation emission matrix (EEM) fluorescence of the water-soluble BrC (WSBrC) and the water-insoluble but methanol-soluble BrC (WI-MSBrC) in the $PM_{2.5}$ using a three-dimensional fluorescence spectrometer. Average light absorption efficiency of both WSBrC ($Abs_{365, WSBrC}$) and WI-MSBrC ($Abs_{365, WI-MSBrC}$) at 365 nm was found to be highest in winter ($10.4\pm6.76$ $Mm^{-1}$ and $10.0\pm5.13$ $Mm^{-1}$, respectively) and distinct from season to season. Averages of fluorescence index (FI) and biological index (BIX) of WSBrC were lower in summer than in other seasons and opposite to that of humification index (HIX), which implied that the secondary formation and further chemical processing of aerosols were intensive during the summer period than in other seasons. Whereas in winter, the higher HIX together with the higher FI and BIX of WI-MSBrC suggested that the BrC loading was mainly influenced by primary emissions from biomass burning and coal combustion. Based on EEM, the types of fluorophores in WSBrC were divided into humic-like substances (HULIS), including low-oxygenated and high-oxygenated species, and protein like compounds (PLOM), whereas mostly PLOM in the WI-MSBrC. The direct radiation absorption by both WSBrC and WI-MSBrC in the range of 300–400 nm was accounted for ~40% to that ($SFE_{Abs}$, $4.97\pm2.71$ $Wg^{-1}$ and $7.58\pm5.75$ $Wg^{-1}$, respectively) in the range, 300–700 nm.

## 1 Introduction

Brown carbon (BrC) is a part of organic aerosol (OA) and can absorb solar radiation in the near-ultraviolet (UV) to visible (Vis) light, ranging from 300-500 nm (Liu et al., 2013). It has been well recognized that BrC has a significant effect on radiative forcing at both regional and global scales (Feng et al., 2013;Jo et al., 2016;Park et al., 2010). *For example*, the warming effect of water-soluble BrC in the Arctic has been reported to be accounted for ~30% of that exerted by the black carbon (Yue et al., 2022). The BrC not only affects the direct radiative forcing, but also has a potential impact on indirect radiative forcing due to its hydrophilicity, which influences the formation of cloud condensation nuclei (CCN) (Andreae and Gelencs´er, 2006;Laskin et al., 2015b). In addition, BrC is mostly composed of highly conjugated aromatic ring compounds such

as a polycyclic aromatic hydrocarbons and high molecular weight substances with a polar
functional group that consists of nitrogen and/or oxygen, or humic-like substances (HULIS), which
could pose a risk to human health. *For example*, carbon-containing aromatic compounds can cause
physical weakness, decreased immunity, arteriosclerosis, etc., which will increase the mortality
due to cardiovascular and cerebrovascular diseases and a variety of cancers such as skin cancer,
pharyngeal cancer and nasal cancer (Diggs et al., 2011;Peters et al., 2008;Hecobian et al., 2010).
BrC can be emitted directly from primary sources such as biomass burning (Hoffer et al.,
2006;Brown et al., 2021), fossil fuel combustion (Jo et al., 2016), and non-combustion processes
such as bioaerosols (plant debris and fungi) and soil humus (Lin et al., 2014;Rizzo et al.,
2013;Rizzo et al., 2011). On the other hand, BrC can also be produced from complex chemical
reactions of volatile organic compounds (VOCs) emitted from both anthropogenic and biological
origin in gas-pahse as well as by multiphase reactions between the gaseous, particulate and
aqueous constituents (Kasthuriarachchi et al., 2020;Li et al., 2020a;Laskin et al., 2015a).
In recent times, after establishing the fact that BrC absorb the light, the researchers are paying
lot of attention to measure the physical (optical) and chemical characteristics of the BrC and
estimate its climatic effects (Yue et al., 2019;Choudhary et al., 2021;Hecobian et al., 2010).
However, studies on BrC are still very limited due to difficulties in quantitative measurement of
light-absorbing organic components (Corbin et al., 2019;Wang et al., 2022b). In fact, based on the
disparity in wavelength dependence between BC and BrC, traditional optical instruments can be
used to obtain the BrC absorption value, but the availability of such instruments are limited, which
can accurately and directly differentiate the light absorption caused by the BC and BrC. On the
other hand, the molecular composition and optical properties of BrC are significantly changed
when the BrC is subjected for physical and photo-chemical processing (aging) in the atmosphere.
That is why, the indirect approaches have been developed to explore the molecular composition
including chromophores and sources of BrC through its light absorption and fluorescence
characteristics.
UV-Vis spectroscopy and excitation emission matrix (EEM) fluorescence spectroscopy are
considered to be common techniques for studying the optical absorption and fluorescence
chromophore optical and structural characteristics of complex organic materials, because each
chromophore has its own specific excitation-emission peak in the EEM maps (Chen et al.,
2016b;Coble, 2007). In recent years, combined spectrophotometric measurement and chemical
analysis has been applied to study the BrC in Xi'an, Northwest China (Huang et al., 2018). In fact,
EEM fluorescence spectroscopy provides multiple superposed spectral data. By using parallel
factor (PARAFAC) analysis of such spectral data, the type of chromophores can be identified and
their types are quantified semi-quantitatively based on the range of excitation-emission
wavelengths (Cao et al., 2022;Zhan et al., 2022;Murphy et al., 2013). The composition of humic-
like and protein-like components have been identified from the analysis of chromophores of
dissolved organic substances in aquatic environments (Xie et al., 2020). The fluorescence
technique has been widely applied to measure organics in terrestrial and oceanic systems (Murphy
et al., 2013;Yu et al., 2015), but has rarely been used in the study of atmospheric aerosols. Now,
the application of fluorescence technique has been well established in studying the molecular
composition of aerosols as well, the studies on identification of chromophores and thus the
molecular composition of BrC in the atmospheric aerosols are still very limited (Wu et al.,
2021a;Deng et al., 2022;Li et al., 2022;Cao et al., 2022).
Therefore, much attention need to be paid further, particularly on long-term and continuous
measurements of the optical characteristics of water-soluble BrC (WSBrC) and their temporal and

spatial variations. Moreover, the investigation of light absorption and fluorescence characteristics of water-insoluble BrC (WIBrC) that can be extracted into a solvent with higher extraction efficiency is necessary to better understand the impact of the BrC on climate change (Corbin et al., 2019). In fact, such studies are very scarce, because the selection of solvents and determination of extraction efficiency are difficult, although different polar chromophores could be extracted by solvent extraction according to the polarity of solvent and methanol has been used as a common solvent (Chen et al., 2016a). Hence, the comprehensive study of the optical properties of WSBrC and WIBrC is highly necessary to better understand the types of chromophores and optical properties of atmospheric aerosols, as well as the processes of oxidation and transformations of chromophores at different locale over the world.

China is one of the most polluted areas in the world, and suffering from the absorption and scattering of solar radiation by atmospheric aerosols that directly affect the energy balance of the Earth's climate system, especially in North China Plain (Wang et al., 2022a). As an important port city in the North China Plain, Tianjin, which has a large population, has received a widespread attention to address the atmospheric environmental issues. Previous studies have shown that BrC in the atmosphere contributes significantly to the light absorption by aerosols (Deng et al., 2022). $PM_{2.5}$ loading in the Tianjin area is extremely high, with greater abundance of organic matter (OM) (Dong et al., 2023a). In such an environment, BrC is likely to become an important light-absorbing component of atmospheric aerosols. However, the studies on physico-chemical characteristics and sources of BrC are very limited in the North China Plain, and to the best of our knowledge, the long-term observations of the optical properties and molecular composition of BrC have not been reported yet over the Tianjin region.

In this study, we measured the optical properties and molecular composition of WSBrC and water-insoluble but methanol-soluble BrC (WI-MSBrC) in fine aerosols ($PM_{2.5}$) collected from Tianjin, North China over a one-year period using the combined UV-Vis absorption and EEM fluorescence spectroscopy technique. We discussed the seasonal variations in optical properties and chromophore composition of WSBrC and WI-MSBrC in the $PM_{2.5}$. We also assessed the possible sources of BrC including the potential photochemical processing of OA (aging) over the Tianjin region, based on the relationships between the BrC and chemical tracers and stable carbon ($\delta^{13}C$) and nitrogen ($\delta^{15}N$) isotope ratios of total carbon (TC) and nitrogen (TN) in the $PM_{2.5}$. Thus, this study provides a comprehensive understanding of the optical characteristics, seasonality, and sources of BrC in the Tianjin region, and warrant the need to develop the prevention and control strategies for the BrC and/or its precursors emissions.

## 2 Materials and Methods

### 2.1 Aerosol sampling

Fine aerosol ($PM_{2.5}$) sampling was conducted in Tianjin, a coastal city located at the lower reaches of the Haihe River and Bohai Sea and 150 km away from Beijing in the northern part of China. The sampling took place on the rooftop of a six-storey building at Tianjin University (ND, 39.11°N,117.18°E) in an urban area of Nankai District, Tianjin. A high-volume air sampler (Tisch Environmental, TE-6070DX) at a flow rate of 1.0 $m^3$ $min^{-1}$ and pre-combusted (6 hours at 450°C) quartz fiber filters (Pallflex 2500QAT-UP) were used for continuously collecting the $PM_{2.5}$ samples for 3 days (~72 hours) each during 5 July 2018 to 4 July 2019 ($n$ = 121). Filter blanks were collected twice per season during the sample campaign, following the same sampling procedure placing the filter in hood for 10 mins without turning on the sampler pump.

Prior to and after sampling, each filter was dehumidified in a desiccator for 48 hours and
determined the PM$_{2.5}$ mass by gravimetric analysis, and then stored in a pre-combusted glass jar
with a Teflon-lined cap in the dark at −20°C until analysis.
2.2 Measurement of carbonaceous and ionic components
Details of the measurements of aerosol organic carbon (OC), element carbon (EC) and water-
soluble organic carbon (WSOC) were described by Wang et al. (Wang et al., 2019) and Dong et
al. (Dong et al., 2023a). Briefly, concentrations of the OC and EC were measured using an aliquot
of filer (1.5. cm$^2$) and a thermal-optical carbon analyzer (Sunset Laboratory Inc, USA), following
the IMPROVE protocol of the protective visual environment. WSOC was measured using an
aliquot of filter (one disc of either 14 mm or 22 mm in diameter) extracted into organic-free Milli
Q water and total organic carbon (TOC) analyzer (Model OI, 1030W + 1088). Concentrations of
K$^+$ and Cl$^−$ were determined using an aliquot of filer (one disc of 22 mm in diameter) extracted
into ultrapure water (>18.2MΩ cm) and ion chromatography (ICS-5000 System, China, Dai An)
(Dong et al., 2023a). The analytical uncertainty in replicate analyses were within 2 % for OC and
5% for EC, WSOC and inorganic ions. Concentrations of all the components were corrected for
field blanks.
2.3 Measurement of optical properties of brown carbon (BrC)
2.3.1 Extraction and concentration of BrC
BrC was extracted into 30 ml ultrapure water using a sample filter disc of 22 mm in diameter
placed in a glass bottle with screw cap and sealed with Teflon tape under ultrasonication for 30
min. The extracts were filtered through a 0.45 μm polytetrafluoron (PTFE) syringe filter to remove
the water-insoluble particles and filter debris, and transferred into a clean glass bottle. The extracts
were used for the light absorption and fluorescence measurements of WSBrC. While the
concentration of WSBrC was considered as the concentration of WSOC.
After the extraction of WSBrC, the WI-MSBrC was extracted into 30 ml methanol using the
same filter sample left in the same glass bottle with screw cap sealed with Teflon tape under
ultrasonication for 30 min. The extracts were filtered using the same 0.45 μm PTFE syringe filter
to remove the insoluble particles and filter debris and transferred into another clean glass bottle.
The methanol extracts were used for the measurements of optical properties of WI-MSBrC. The
concentration of water-insoluble organic carbon (WIOC) was considered as the concentration of
WI-MSBrC, which calculated as equation (1), presuming that all the water-insoluble organic
contents are dissolved in methanol, although we do not preclude that some of organic species are
not soluble in MeOH (Shetty et al., 2019).
$WI - MSBrC = OC - WSOC$                   (1)
2.3.2 Light absorption of BrC
A three-dimensional fluorescence spectrometer (Aqualog, Horiba Scientific) was used to
record the excitation-emission matrices (EEM) spectra and ultraviolet-visible (UV–Vis)
absorption spectra of the solution samples in 1×1 cm quartz cuvettes. The instrument parameters
during sample analysis were as follows: The UV-Vis absorption spectra of extracts were recorded
in the wavelength range of 240–700 nm. The UV−visible absorption spectra of the solvents were
also recorded to subtract their contributions from the extract spectra. The EEM was recorded in
the wavelength range of 240–700 nm for excitation and the integration time was 0.1 s with a 1 nm
increment. An increment of 8 pixels (5.04 nm) is used as the emission wavelength interval. Prior
to sample analysis, the pure solvents of water and methanol (MeOH) were used to obtain the
reference signal.

Based on the light absorption spectra, the absorption data are converted to the absorption
coefficient (Abs: $m^{-1}$) following equation (2) (Deng et al., 2022;Hecobian et al., 2010):
$$Abs_\lambda = (A_\lambda - A_{700}) \times \frac{V_l}{V_a} \times ln(10) \tag{2}$$
where $A_{700}$ is the absorption at 700 nm, serving as a reference to account for baseline drift; $V_l$ is
the volume of water or MeOH used for extraction; $V_a$ is the volume of sampled air; L is the optical
path length (0.01 m). A factor of ln(10) is utilized to convert the log base 10 to a natural logarithm
to obtain a base-e absorption coefficient. To compensate for any baseline shift that may occur
during analysis, absorption at wavelengths below 700 nm is compared to that of 700 nm where no
absorption occurs for ambient aerosol extracts. The average absorption coefficient between 360
and 370 nm ($Abs_{365}$) is used to represent BrC absorption in order to avoid any interferences from
non-organic compounds (e.g., nitrate) and to be consistent with the literature values (Huang et al.,
2018).

Absorption Ångström exponent (AAE, Å) represents the spectral dependence of aerosol light
absorption. The spectral dependence of light absorption by chromophores in solution can be
described by the following equation (3):
$$Abs_\lambda = C \times \lambda^{-AAE} \tag{3}$$
where C is a composition-dependent constant; λ is the wavelength (nm). The AAE of the filter
extracts is calculated by a formula in the wavelength range of 300–500 nm. The selected range
serves two purposes: (1) to prevent any interferences from non-organic compounds at lower
wavelengths; (2) to ensure a sufficiented signal-noise ratio for the investigating samples (Huang
et al., 2018).

The mass absorption efficiency (MAE: $m^2\,g^{-1}$) of the filter extract at wavelength of λ can be
characterized as equation (4). The ratio of $MAE_{250}$ to $MAE_{365}$ is denoted as $E_2/E_3$ to characterize
the relative size of molecular weight, which is inversely proportional to the molecular weight.
$E_2/E_3$ is calculated with the method as equation (5).
$$MAE_\lambda = Abs_\lambda / M \tag{4}$$
$$\frac{E_2}{E_3} = \frac{MAE_{250}}{MAE_{365}} \tag{5}$$
where M ($\mu g\,m^{-3}$) is the concentration of WSOC for water extracts and that of WIOC for methanol
extracts.

The imaginary part ($k$) of the refractive index (m = n+i$k$) is derived with the following
equation (6) (Liu et al., 2013;Deng et al., 2022):
$$k_\lambda = (MAE \times \rho \times \lambda)/4\pi \tag{6}$$
where MAE is the mass-absorption cross section of WSBrC or WI-MSBrC ($m^2\,g^{-1}$), ρ is the
effective density, λ is the wavelength for the computed MAE including WSBrC and WI-MSBrC.
For this study, an effective density of 1.5 g $m^{-3}$ is assumed for WSBrC and WI-MSBrC in the
derivation (Liu et al., 2013). MAE values are computed for 365 nm.
2.3.2 EEM of BrC and PARAFAC analysis

The raw EEMs were first calibrated for the correction of spectrometer factors, which reflect
the spectrometer deviation and light source, and then for the inner filter correction, following the
procedure described elsewhere (Chen et al., 2019;Gu and Kenny, 2009). Briefly, the inner filter

correction of the EEMs was done based on the UV-Vis light absorbance of the extracts, which was lower than 0.7 in the calibrated wavelength range and is appropriate (Gu and Kenny, 2009). The signal intensity of the EEMs was then normalized to the Raman unit (RU) of water (Lawaetz and Stedmon, 2009). The fluorescence volume (FV, RU-nm$^2$/m$^3$) of extracts present in the atmosphere was estimated based on the EEMs at the excitation wavelength ranging from 240 to 700 nm, and then normalized it (i.e., NFV (RU-nm$^2$-[mg/L]$^{-1}$)) by dividing the FV with the concentration of WSOC and WIOC in the aerosol [mg m$^{-3}$]).

Various types of chromophores present in the PM$_{2.5}$ samples were classified and identified based on the PARAFAC analysis of the EEMs using the SOLO (Eigenvector Inc.), the data analysis software. PARAFAC analysis was performed for each extraction fluid in each season. Ultimately, three EEM components were determined and assigned to different types of chromophores.

Additionally, fluorescence index (FI) was determined by calculating the ratio of emission intensities at 450 nm and 500 nm after excitation at 370 nm (McKnight et al., 2001). Contributions from local biological sources can be characterized by biological index (BIX), which was calculated using the ratio of emission intensities at 380 and 430 nm following 310 nm excitation (Gao yan and Zhang, 2018). Under the condition of Ex=255 nm, the humification index (HIX) was determined by dividing the area of fluorescence intensity between 435 and 480 nm by that of fluorescence intensity between 300 and 345 nm (Battin, 1998). The calculation formulas (7)– 9) are as follows:

$$FI = \frac{F_{450}}{F_{500}}, \lambda_{Ex} = 370nm \tag{7}$$

$$BIX = \frac{F_{380}}{F_{430}}, \lambda_{Ex} = 310nm \tag{8}$$

$$HIX = \frac{\int 435-480}{\int 300-345}, \lambda_{Ex} = 255nm \tag{9}$$

In formula (6) - (8), $\lambda_{Ex}$ refers to the excitation wavelength, $F_i$ refers to the fluorescence intensity of emission wavelength at i in the emission spectrum, and $\int$i-j refers to the integrated fluorescence emission intensity in the range of 435–480 nm to 300–345 nm.

### 2.3.3 Simple forcing efficiency by light absorption of BrC (SFE$_{Abs}$)

It is possible to make a rough estimate of the radiative forcing caused by aerosols using a simple forcing efficiency (SFE, W/g), which reflects the energy added to the Earth's atmospheric system per unit mass of aerosols and can be estimated as described in the literature (Bond and Bergstrom, 2006;Deng et al., 2022), using the following equation (10):

$$\frac{dSFE}{d\lambda} = -\frac{1}{4}\frac{dS(\lambda)}{d\lambda}\tau_{atm}^2(\lambda)(1-F_c)[2(1-a_s)^2\beta(\lambda) \times MSE(\lambda) - 4a_s \times MAE(\lambda)] \tag{10}$$

where dS/dλ is the solar irradiance, $\tau_{atm}$ is the atmospheric transmission (0.79), $F_c$ is the cloud fraction (approximately 0.6), a is the surface albedo (average 0.19), β is the backscatter fraction, and MSE and MAE are the mass scattering and absorption efficiency, respectively (Deng et al., 2022).

It is important to note that BrC can affect a strong instantaneous negative forcing through scattering, however it is not possible to evaluate this directly using offline samples because of strong dependency on particle size. That is why, we limited to estimate the radiative effect caused by only the absorption component of the BrC in this study. Therefore, the equation (10) can be simplified to:

$$SFE_{Abs} = \int \frac{dS(\lambda)}{d\lambda} \tau_{atm}^2 (1 - F_c)a_s MAE(\lambda)d\lambda \tag{11}$$

**3 Results and discussion**

3.1 Characteristics of ultraviolet light absorption of WSBrC and WI-MSBrC

3.1.1 Absorption coefficient (Abs)

Annual and seasonal averages of various optical properties of WSBrC and WI-MSBrC in PM$_{2.5}$ measured in this study are summarized in Table 1. Their ranges and median values are provided in supplement Table S1. Temporal variations in absorption coefficient of WSBrC at 365 nm (Abs$_{365(WSBrC)}$) and that of WI-MSBrC (Abs$_{365(WI-MSBrC)}$) together with the concentrations of WSOC and WIOC are depicted in Fig. 1. Because the light absorption at the wavelength of 365 nm would not be interfered by inorganic substances (Hecobian et al., 2010), the Abs at 365 nm was selected for the analysis in this study. Abs$_{365(WSBrC)}$) ranged from 0.49 Mm$^{-1}$ to 36.7 Mm$^{-1}$ with an average of 4.74 Mm$^{-1}$ during the campaign. While the Abs$_{365(WI-MSBrC)}$ ranged from 0.32-25.0 Mm$^{-1}$ (avg. 3.87 Mm$^{-1}$) during the campaign. Temporal trends of Abs$_{365(WSBrC)}$ were found to be similar with those of Abs$_{365(WI-MSBrC)}$, with the lowest levels in summer followed by a gradual increase toward autumn and peak in winter and then a gradual decrease toward spring during the campaign (Fig. 1). Furthermore, those trends were highly comparable to those of the concentrations of both WSOC and WIOC in PM$_{2.5}$ (Fig. 1). The correlations between Abs$_{365(WSBrC)}$ and WSOC and Abs$_{365(WI-MSBrC)}$ and WIOC were found to be strong (R = 0.93 and 0.96, respectively) during the campaign. These results indicate that both WSBrC and WI-MSBrC might have been derived from the same or similar sources including the secondary processes, and their light absorbance should have been significantly dependent on their abundances that varied from season to season (Fig. 1; Table 1).

Averages of both Abs$_{365(WSBrC)}$ and Abs$_{365(WI-MSBrC)}$ were higher in winter followed by autumn and spring and the lowest in summer (Table 1). The high Abs$_{365}$ of BrC in winter might have been mainly driven by the existence of large amounts of organic aerosols, whereas the lowest Abs$_{365}$ in summer might be due to enhanced decomposition of BrC constituents by photobleaching under high solar radiation and oxidants loading in the atmosphere, which is unlikely in the wintertime. The seasonal variations of both Abs$_{365(WSBrC)}$ and Abs$_{365(WI-MSBrC)}$ in Tianjin were similar to those of the Abs$_{365}$ of WSBrC reported in the southeastern United States, but their values (Table 1) were much higher than that (0.3–3.0 Mm$^{-1}$ in 2007) in the southeastern United States (Hecobian et al., 2010) as well as that in Atlanta and Los Angeles ($0.88 \pm 0.71$ and $0.61 \pm 0.38$ Mm$^{-1}$, respectively) in summer 2010 (Zhang et al., 2011). Biomass burning was considered to be the dominant source of BrC at the southeastern United States in colder period, whereas both primary emissions from fossil fuel combustion and secondary formation were significant in summertime (Hecobian et al., 2010). While the SOA formed from fresh anthropogenic and biogenic VOCs were considered to be major at Atlanta and Los Angeles, respectively (Zhang et al., 2011).

It has been reported that the solid fules (i.e., biomass or coal) combustion is dominant and the Abs$_{370}$ of BrC is reported to be high (21.8 Mm$^{-1}$) in North China cities (Zhang et al., 2021). It has also been reported that the Abs$_{370}$ of BrC produced by residential wood burning is much higher, reaching up to $37.1 \pm 74.6$ Mm$^{-1}$ in Athens in winter (Liakakou et al., 2020). The maximum Abs$_{365(WSBrC)}$ and Abs$_{365(WI-MSBrC)}$ in Tianjin aerosols were 36.7 and 25.0 Mm$^{-1}$, respectively, which are comparable to those of wood combustion samples. However, their ranges found to be

large during the campaign (Fig. 1; Table S1), suggesting that in addition to biomass burning, the other emission sources and meteorological conditions in different seasons should have been played an important role in controlling the WSBrC and WI-MSBrC loadings and their optical characteristics in the Tianjin atmosphere. Furthermore, the $Abs_{365(WSBrC)}$ observed in this study (Table 1) is slightly lower compared to that reported in Tianjin during winter 2016 ($14.1 \pm 8.5$ $Mm^{-1}$) and summer 2017 ($2.1 \pm 1.0 Mm^{-1}$) (Deng et al., 2022) as well as that reported in Beijing and Xi'an, which are considered to be highly polluted cities in northern China (Huang et al., 2020;Li et al., 2020b). However, the $Abs_{365(WSBrC)}$ and $Abs_{365(WI-MSBrC)}$ found in winter in this study were higher than that reported at different locations in southern China; Nanjing ($Abs_{365(WSBrC)} =$ 4.84 $Mm^{-1}$, $Abs_{365(MSBrC)} = 7.75$ $Mm^{-1}$) (Xie et al., 2020), Guangzhou ($Abs_{365(WSBrC)} = 8.8$ $Mm^{-1}$) (Li et al., 2018), and Lhasa ($Abs_{365(WSBrC)} = 1.04$ $Mm^{-1}$, $Abs_{365(MSBrC)} = 1.47$ $Mm^{-1}$) (Zhu et al., 2018), where the fossil fuel combustion is considered as the dominant source. Such higher $Abs_{365}$, particularly in winter, indicates that BrC in $PM_{2.5}$ in Tianjin might have been derived from mixed sources such as biomass burning and fossil fuel (coal) combustion and has a significant effect on light absorption and thus on climate system over the region.

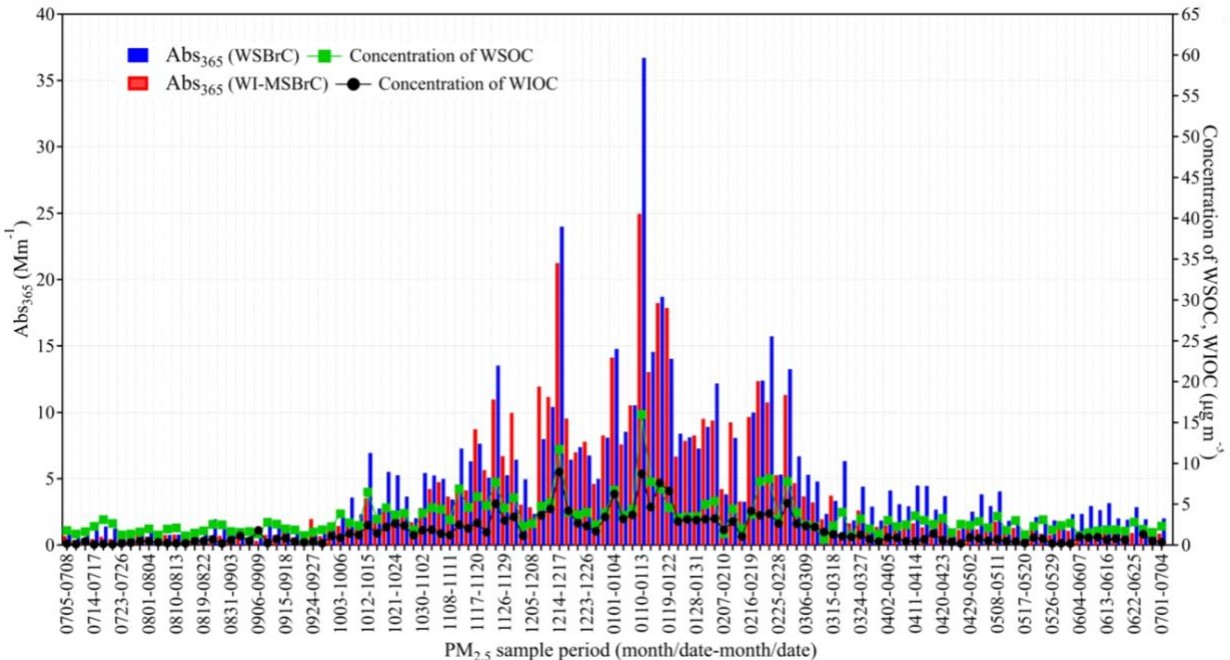

**Figure 1**. Temporal variations of the light absorption coefficient of water-soluble brown carbon (BrC) at 365 nm ($Abs_{365(WSBrC)}$) and water-insoluble but methanol-soluble BrC ($Abs_{365(WI-MSBrC)}$) and the mass concentrations of WSOC and WIOC in $PM_{2.5}$ in Tianjin, North China during 2018 and 2019. WSOC and WIOC mass concentrations data was obtained from (Dong et al., 2023b).

Figure 2 shows the seasonal average absorption spectra of WSBrC and WI-MSBrC at wavelengths of 240−700 nm, which shows a common feature that the absorption of shorter wavelengths increases sharply and significantly. Such feature is different from the absorption characteristics of BC, whose AAE is close to 1 and weakly dependent on the wavelength. Another evident feature of BrC absorption spectra shown in Figure 2 is that the Abs of WI-MSBrC was always greater than that of WSBrC across the shorter wavelengths in winter and in the range of 260~300 nm in other seasons, which is consistent with the pattern reported in the literature (Huang

et al., 2020;Li et al., 2020b). In addition, the Abs of WI-MSBrC peaked at 280 nm, but not that of
WSBrC (Fig. 2). Such patterns can be attributed to the difference in types and amounts of
chromophores soluble in water and methanol (e.g., PAHs are soluble in methanol, but not in water).
It is noteworthy that, $\pi-\pi*$ electron transitions in the double bonds of aromatic compounds are the
primary cause of light absorption in the wavelength range of 250−300 nm. It has been reported in
another study that nitroaromatics have contributed 60% to the total absorbance in the 300-400 nm
range (Hems et al., 2021). The electron transitions in phenolic arenes, aniline derivatives, polyenes
and polycyclic aromatic hydrocarbons with two or more rings are responsible for the absorbance
in the bands between 270 and 280 nm (Baduel et al., 2009). Therefore, the differences observed in
the Abs of WSBrC and WI-MSBrC imply that the aromatic and/or unsaturated aliphatic organic
compounds are abundant in $PM_{2.5}$ in Tianjin, which are more soluble in MeOH than in water.
High correlations (R = 0.73-0.97) were found between $Abs_{365}$ of both WSBrC and WI-
MSBrC and WSOC and WIOC in each season, except in summer (R = 0.20-0.62) (Figure S1). As
noted earlier, such linearity of $Abs_{365}$ with WSOC and WIOC indicate that WSBrC and WI-
MSBrC might have been derived from similar sources including the secondary processes over the
Tianjin region, except in summer, because the light absorption efficiency of organic compounds
of different origin are different and significantly depend on their secondary processes in the
atmosphere (Zhong and Jang, 2011). In fact, the Abs depends on the amount of BrC availability,
but not of total OC content. In summer, the BrC loading might be less due to either photobleaching
under the enhanced aging and/or less availability of N and/or S species to produce N- and S-
containing organics (BrC) in the atmosphere over the Tianjin region.

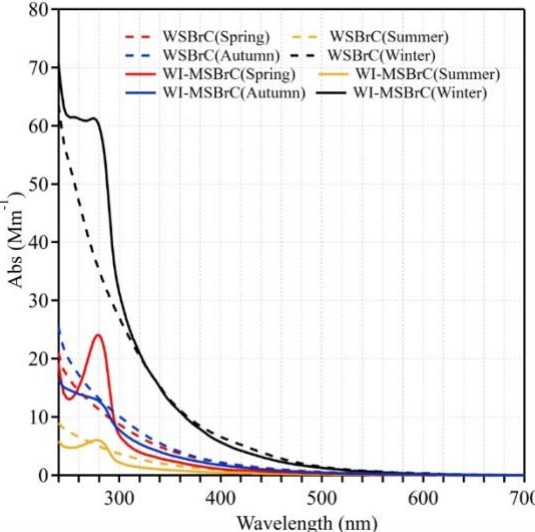


**Figure 2**. Seasonal averages of absorption spectra in the wavelength range of 240–700 nm of
WSBrC and WI-MSBrC in $PM_{2.5}$ from Tianjin, North China.
The moderate to high positive correlations (R = 0.51-0.92) found between both $Abs_{365(WSBrC)}$
and $Abs_{365(WI-MSBrC)}$ and $K^+$ and $Cl^-$ in all seasons, except between $Abs_{365\,WSBrC}$ and $Cl^-$ in summer
(R = 0.29) (Fig. S2), suggest that biomass burning and coal combustion were major sources (Dong
et al., 2023a) of BrC in the Tianjin region. The poor correlation between $Abs_{365}$ and $K^+$ was driven
by two outliers obtained in $K^+$ data that might have occurred due to unknown biomass burning
events at local scale. In addition, the correlation between $Abs_{365(WSBrC)}$ and $K^+$ was relatively
stronger than that between the $Abs_{365(WI-MSBrC)}$ and $K^+$, except in summer (Fig. S2), which support

that the chromophores, like nitrophenols, derived from biomass burning are potentially more water-soluble (Li et al., 2020b). While the correlation between $Abs_{365(WI\text{-}MSBrC)}$ and $Cl^-$ was relatively stronger than that between $Abs_{365(WSBrC)}$ and $Cl^-$ in spring and summer and comparable in autumn, which suggest that the chromophores derived from fossil fuel (e.g., coal) combustion are slightly more soluble in MeOH compared to that in water, and were abundant in the spring and summer time in Tianjin.

**Table 1.** Mass concentrations of WSOC, WIOC and absorbance efficiency of WSBrC and WI-MSBrC (Avg. ± SD) in $PM_{2.5}$ from Tianjin, North China.

| | | Annual | Summer | Autumn | Winter | Spring |
|---|---|---|---|---|---|---|
| **Concentrations** | | | | | | |
| WSOC (μg m$^{-3}$) | | 3.25 ± 2.18 | 1.88±0.53 | 3.45±1.71 | 5.06±2.99 | 2.48±0.82 |
| WIOC (μg m$^{-3}$) | | 1.68 ± 1.77 | 0.43±0.32 | 1.55±1.04 | 3.74±2.09 | 0.88 ± 0.63 |
| **Optical parameters** | | | | | | |
| WSBrC | $Abs_{365}$(Mm$^{-1}$) | 4.74±5.10 | 1.47±0.77 | 3.71±2.83 | 10.4±6.76 | 3.45±2.29 |
| | $MAE_{365}$(m$^2$ g$^{-1}$) | 1.28±0.66 | 0.80±0.44 | 0.96±0.33 | 2.04±0.46 | 1.31±0.55 |
| | AAE (300–500 nm) | 5.66±0.82 | 5.17±0.83 | 6.21±0.65 | 5.88±0.58 | 5.42±0.74 |
| | $E_2/E_3$ | 5.36±0.91 | 5.64±1.21 | 5.78±0.83 | 5.19±0.44 | 4.83±0.64 |
| | FI | 1.38±0.09 | 1.31±0.07 | 1.47±0.07 | 1.37±0.02 | 1.37±0.09 |
| | BIX | 1.05±0.13 | 0.91±0.06 | 1.06±0.08 | 1.20±0.08 | 1.01±0.11 |
| | HIX | 2.87±0.53 | 3.12±0.44 | 3.11±0.51 | 2.47±0.43 | 2.76±0.47 |
| | $k_{365}$ | 0.056±0.029 | 0.035±0.020 | 0.042±0.015 | 0.089±0.021 | 0.057±0.024 |
| | $SFE_{Abs300-400}$ (w g$^{-1}$) | 1.95±1.02 | 1.21±0.67 | 1.99±0.84 | 3.12±0.71 | 1.46±0.52 |
| | $SFE_{Abs300-700}$ (w g$^{-1}$) | 4.97±2.71 | 3.68±2.58 | 5.12±2.17 | 7.60±2.17 | 3.39±1.42 |
| WI-MSBrC | $Abs_{365}$(Mm$^{-1}$) | 3.87±4.69 | 0.74±0.25 | 2.83±2.51 | 10.0±5.13 | 1.99±1.95 |
| | $MAE_{365}$(m$^2$ g$^{-1}$) | 2.36±1.26 | 2.50±1.78 | 1.86±1.02 | 2.69±0.36 | 2.41±1.28 |
| | AAE (300–500 nm) | 6.06±1.23 | 5.49±1.26 | 6.11±1.86 | 6.30±0.27 | 6.27±0.90 |
| | $E_2/E_3$ | 6.60±2.04 | 6.79±1.32 | 5.77±1.35 | 6.20±0.44 | 7.60±3.25 |
| | FI | 1.60±0.13 | 1.58±0.12 | 1.57±0.06 | 1.73±0.11 | 1.51±0.11 |
| | BIX | 1.26±0.21 | 1.32±0.18 | 1.05±0.14 | 1.43±0.09 | 1.23±0.18 |
| | HIX | 0.81±0.60 | 0.25±0.08 | 1.23±0.61 | 1.33±0.30 | 0.42±0.28 |
| | $k_{365}$ | 0.104±0.057 | 0.109±0.079 | 0.081±0.045 | 0.117±0.016 | 0.105±0.057 |
| | $SFE_{Abs300-400}$ (w g$^{-1}$) | 2.98±1.70 | 1.21±0.67 | 2.98±1.52 | 4.13±0.57 | 3.61±1.91 |
| | $SFE_{Abs300-700}$ (w g$^{-1}$) | 7.58±5.75 | 3.68±2.58 | 8.69±9.23 | 9.36±4.51 | 8.70±5.03 |

### 3.1.2 Absorption Ångström exponent (AAE)

The magnitude of the AAE can reflect the sources and atmospheric chemical processes of BrC (Lack et al., 2013), because the AAE of the BrC emitted from fossil fuel combustion found to be ~1 and that from biomass burning range from 1 to 3 and that derived by secondary formation/transformations vary from 3-7 (Yan et al., 2018). It has also been reported that the AAE of light-absorbing organic species (i.e., BrC) is much larger than that of soot (BC). The AAE was found to be between 2 and 4 for the particles containing both soot and BrC. Furthermore, AAE value of particulate matter is closely related to its chemical composition, mixing state, particle size and other factors. *For example*, Sun et al. (2007) reported that the average AAE of coal briquettes is $2.55 \pm 0.44$ whereas that of the coal chunks is $1.30 \pm 0.32$ (Sun et al., 2017). However, it is important to note that unlike the direct measurement of AAE of the particulate matter, the light absorption characteristics of organic components extracted into solvent are not affected by particle size and mixing state of aerosols, but depend on their composition. The AAE of humic-like substances (HULIS) isolated from biomass burning aerosols by water extraction followed by the separation with exchange column was reported to be 6-7 (Hoffer et al., 2006).

The AAE of WSBrC in $PM_{2.5}$ from Tianjin ranged from 3.85 to 7.99 with an average of 5.66 during the campaign. The seasonal averages were highly comparable with each other, except a little higher level in autumn (Table 1). The average AAE of WSBrC in Tianjin (Table 1) is comparable to that ($5.1 \pm 2.0$) reported from New Delhi, India and Beijing ($5.3 \pm 0.4$ in winter and $5.8 \pm 0.5$ in summer) and the outflow region ($6.4 \pm 0.6$) of northern China (Lesworth et al., 2010). The AAE of WSBrC in Tianjin was also similar to that (range, 6–8) reported in the particulate matter at the southeastern United States (Hecobian et al., 2010) and downtown Atlanta (Liu et al., 2013), where both biogenic and fossil fuel combustion emissions and secondary processes are considered as significant sources. Such higher levels and comparisons of the AAE of WSBrC imply that the OA in Tianjin should have been derived from mixed sources and substantially polar, because the AAE of BrC is increased with its increasing polarity (Chen et al., 2016a).

However, the AAE of WI-MSBrC in Tianjin ranged from 2.08–12.9 (avg. 6.06) and was comparable with that of WSBrC. Furthermore, the averages of AAE of WI-MSBrC in each season were comparable with the other, except a relatively lower level in summer, and also with those of the AAE of WSBrC (Table 1). Generally, the water insoluble portion is expected to have a stronger absorption and weaker wavelength dependence (Saleh, 2020). It has also been reported that the AAE values of the water extract are greater than those of the acetone and methanol extracts (Shetty et al., 2019), and interpreted that the extraction efficiency of polycyclic aromatic hydrocarbons from methanol or other organic solvents is higher than that from water, leading to a higher absorption at longer wavelengths in the methanol extract and therefore a lower AAE value. However, it has also been found that the value of $AAE_{300-600}$ of water extract of biomass burning samples is lower than that extracted into acetonitrile (Lin et al., 2017), indicating that the origin of the BrC is also play an important role. Such comparability between the AAE of WSBrC and WI-MSBrC is consistent with the pattern reported in urban Beijing during winter and Xi'an, China (Li et al., 2020b), where the emissions from fossil fuel combustion are dominant. These results and their comparisons again support that the BrC might have significantly derived from mixed sources (biomass burning and fossil fuel combustion).

3.1.3 Mass absorption efficiency (MAE) and imaginary refractive index ($k$)
MAE provides the light absorbing ability of BrC. The $MAE_{365}$ of WSBrC ($MAE_{365(WSBrC)}$)
ranged from 0.38 $m^2\,g^{-1}$ to 3.41 (avg. 1.28 $m^2\,g^{-1}$) and lower by 2 times than that (range, 0.18-7.05
$m^2\,g^{-1}$; avg. 2.36 $m^2\,g^{-1}$) of WI-MSBrC ($MAE_{365(WI\text{-}MSBrC)}$) during the campaign in Tianjin.
Although the seasonal averages of both $MAE_{365(WSBrC)}$ and $MAE_{365(WI\text{-}MSBrC)}$ were higher in winter
(1.28 and 2.36 $m^2g^{-1}$, respectively), the former showed the second most value in spring followed
by autumn and the lowest value in summer, whereas the latter showed second most value in
summer followed by spring and the lowest value in autumn (Table 1). Furthermore, the average
$MAE_{365(WSBrC)}$ in winter was 2.5 times higher than that in summer, which is similar to that (1.8
times) reported earlier in Tianjin (Deng et al., 2022), whereas the difference between the averages
of $MAE_{365(WI\text{-}MSBrC}$ in winter to autumn is 1.4 times only. The seasonal variations of $MAE_{365(WSBrC)}$
and $MAE_{365(WI\text{-}MSBrC)}$ found in this study are similar to those reported in Xi'an (Li et al., 2020b).
The imaginary refractive index ($k$) is another important parameter that represent the light
absorbing ability of carbon and applied in climate model to assess the direct radiative forcing of
aerosols. The $k$ of WSBrC ($k_{365(WSBrC)}$) and WI-MSBrC ($k_{365(WI\text{-}MSBrC)}$) in Tianjin ranged from
0.017 to 0.149 and 0.008-0.307, respectively, in Tianjin. Interestingly, the average $k_{365(WI\text{-}MSBrC)}$
was 1.9 times to that of $k_{365(WSBrC)}$ during the campaign (Table 1) and their seasonal patterns were
also exactly similar to those of the $MAE_{365(WSBrC)}$ and $MAE_{365(WI\text{-}MSBrC)}$ (Table 1).
Both these $MAE_{365}$ and $k_{365}$ results indicate that most of light-absorbing chromophores are
insoluble in water but soluble in MeOH, and their abundances are significantly varied from season
to season. Such large seasonal differences indicate that the BrC sources and formation and/or
transformation including the degradation (photobleaching) processes might be different in each
season. The higher levels of $MAE_{365}$ and $k_{365}$ in winter suggest that the contributions of OA from
coal combustion and biomass burning emissions were significantly higher than that in other
seasons due to increased residential heating activities. The lower $MAE_{365(WSBrC)}$ and $k_{365(WSBrC)}$ and
the second most values of $MAE_{365(WI\text{-}MSBrC)}$ and $k_{365(WI\text{-}MSBrC)}$ in summer imply that the
contributions of OA from fossil fuel combustion emissions might be dominant and the subsequent
photobleaching of WSBrC might be significant under high solar radiation in the summertime.
The ratio of $MAE_{250}$ to $MAE_{365}$, which is inversely correlate with the molecular size and
aromaticity (Chen et al., 2016c), of WSBrC ($E_2/E_{3(WSBrC)}$) and WI-MSBrC ($E_2/E_{3(WI\text{-}MSBrC)}$) in
Tianjin ranged from 3.30 to 6.25 with an average of 4.83 and 4.50-24.1 (avg. 7.61), respectively,
during the campaign. Interestingly, the averages of $E_2/E_{3(WSBrC)}$ were comparable in summer and
autumn and higher than that in winter and spring (Table 1). Whereas the average $E_2/E_{3(WI\text{-}MSBrC)}$
was higher in spring followed by summer and winter and the lowest in autumn, and higher than
the $E_2/E_{3(WSBrC)}$ in each season, except in autumn. Both $E_2/E_{3(WSBrC)}$ and $E_2/E_{3(WI\text{-}MSBrC)}$ in each
season were comparable or relatively higher than the $E_2/E_3$ of HULIS (4.7 $\pm$ 0.27 for herbaceous
plants, 3.6 $\pm$ 0.18 for shrubs, 4.2 $\pm$ 0.77 for evergreen trees, 4.0 $\pm$ 0.82 for deciduous trees, 5.8 $\pm$
0.5 for rice straw, 4.5 $\pm$ 0.2 for corn straw and 4.4 $\pm$ 0.3 for pine branches) emitted from biomass
burning (Tang et al., 2020) and lower than that (14.7 $\pm$ 0.7) of HULIS emitted from coal
combustion (Fan et al., 2016). Thus, the $E_2/E_{3(WSBrC)}$ and $E_2/E_{3(WI\text{-}MSBrC)}$ and their comparisons
with source signatures indicate that both WSBrC and WI-MSBrC in $PM_{2.5}$ over the Tianjin region
should have been mainly derived from biomass burning followed by coal combustion and consist
of high aromaticity and large in molecular size.
3.2 Direct radiative forcing of WSBrC and WI-MSBrC
Radiative forcing efficiency of WSBrC and WI-MSBrC were calculated by integrating the
wavelength dependent $SFE_{Abs}$ from 300 nm to 700 nm ($SFE_{Abs300-700(WSBrC)}$ and $SFE_{Abs300-700(WI-}$
$_{MSBrC)}$, respectively) in this study. The $SFE_{Abs300-400}$ was also integrated to estimate the radiative
forcing efficiency of WSBrC ($SFE_{Abs300-400(WSBrC)}$) and WI-MSBrC ($SFE_{Abs300-400(WI-MSBrC)}$),
because the BrC strongly absorbs light in the UV-Vis range. The temporal variations of $SFE_{Abs}$ of
WSBrC and WI-MSBrC in both the wavelength ranges are shown in Fig. 3. $SFE_{Abs300-700(WSBrC)}$
and $SFE_{Abs300-400(WSBrC)}$ ranged from 0.98 $Wg^{-1}$ to 13.1 $Wg^{-1}$ with an average of 4.97 $Wg^{-1}$ and
0.60-5.13 $Wg^{-1}$ (avg. 1.95 $Wg^{-1}$), respectively. Whereas the $SFE_{Abs300-700(WI-MSBrC)}$ and $SFE_{Abs300-}$
$_{400(WI-MSBrC)}$ were 0.92-51.3 $Wg^{-1}$ (7.58 $Wg^{-1}$) and 0.64-8.84 $Wg^{-1}$ (2.98 $Wg^{-1}$), respectively, and
were higher by 1.5 times than that of the $SFE_{Abs300-700(WSBrC)}$ and $SFE_{Abs300-400(WSBrC)}$ (Table 1).
Further both integrated average $SFE_{Abs300-700(WSBrC)}$ and $SFE_{Abs300-700(WI-MSBrC)}$ were higher by 2.5
times to that of the $SFE_{Abs300-400(WSBrC)}$ and $SFE_{Abs300-400(WI-MSBrC)}$ (Table 1). Temporal variations
of both the $SFE_{Abs300-400(WSBrC)}$ and $SFE_{Abs300-700(WSBrC)}$ were found to be quite similar with a clear
seasonal pattern with the lowest levels in summer followed by a gradual increase toward autumn
to peak in winter and then a gradual decrease toward spring to the lowest levels in summer, except
a sharp rise in early summer 2019 (Fig. 3). Whereas the $SFE_{Abs300-400(WI-MSBrC)}$ and $SFE_{Abs300-700(WI-}$
$_{MSBrC)}$ showed exactly the similar temporal pattern with each other, but different from that of the
$SFE_{Abs300-400(WSBrC)}$ and $SFE_{Abs300-700(WSBrC)}$ (Fig.3). The levels of $SFE_{Abs300-400(WI-MSBrC)}$ and
$SFE_{Abs300-700(WI-MSBrC)}$ found to be relatively stable throughout each season, except in spring, with
higher level in spring followed by winter and lower levels in summer (Fig. 3). In consistent with
these seasonal patterns, the seasonal variations of $k_{365(WSBrC)}$ and $k_{365(WI-MSBrC)}$, a vital parameter
that reflect the light absorbing ability and used in the estimation of radiative forcing by climatic
model (Shamjad et al., 2016), were also showed the similar pattern (Fig. S3).
The $SFE_{Abs}$ of both WSBrC and WI-MSBrC in both the spectral ranges were higher in winter
(Table 1). However, $SFE_{Abs300-400(WSBrC)}$ and $SFE_{Abs300-700(WSBrC)}$ showed the second higher values
in autumn and the lowest and comparable values in summer and spring (Table 1). Whereas
$SFE_{Abs300-400(WI-MSBrC)}$ and $SFE_{Abs300-700(WI-MSBrC)}$ showed the second higher and comparable values
in spring and autumn and the lowest values in summer (Table 1). It is noteworthy that the
$SFE_{Abs300-400(WSBrC)}$, $SFE_{Abs300-700(WSBrC)}$, $SFE_{Abs300-400(WI-MSBrC)}$ and $SFE_{Abs300-700(WSBrC)}$ were
higher by 61%, 52%, 71% and 61%, respectively, in winter than those in summer, indicating that
BrC abundance and strong light absorption capacity of BrC in winter led to a significant increase
in direct radiative forcing by the BrC. Furthermore, $SFE_{Abs300-400}$ accounted for 40% of $SFE_{Abs300-}$
$_{700}$ in both the fractions of BrC during the whole campaign period and their seasonal averages
varied between 33-44%, which are similar to that reported in Tianjin by Deng et al. (2022),
indicating the light absorption by BrC in UV-Vis range play a significant role in the total BrC
radiative forcing.
Furthermore, it is important to note that it has been reported that direct radiative effect of
WSBrC is 12.5%% and 13.5% relative to black carbon (BC) radiative forcing in the 280-4000 nm
range in summer and winter, respectively, in Tianjin (Deng et al., 2022). In fact, as noted above,
the annual average $SFE_{Abs300-700(WI-MSBrC)}$ is higher by 1.5 times to that of $SFE_{Abs300-700(WSBrC)}$
(Table 1) in Tianjin. Therefore, the direct radiative effect of total ($\sum$WSBrC+WI-MSBrC) BrC
relative to BC would become ~32.5% in Tianjin, revealing that the BrC play a greater role in light
absorbing aerosols in the shorter wavelength region in comparison to the entire spectrum.

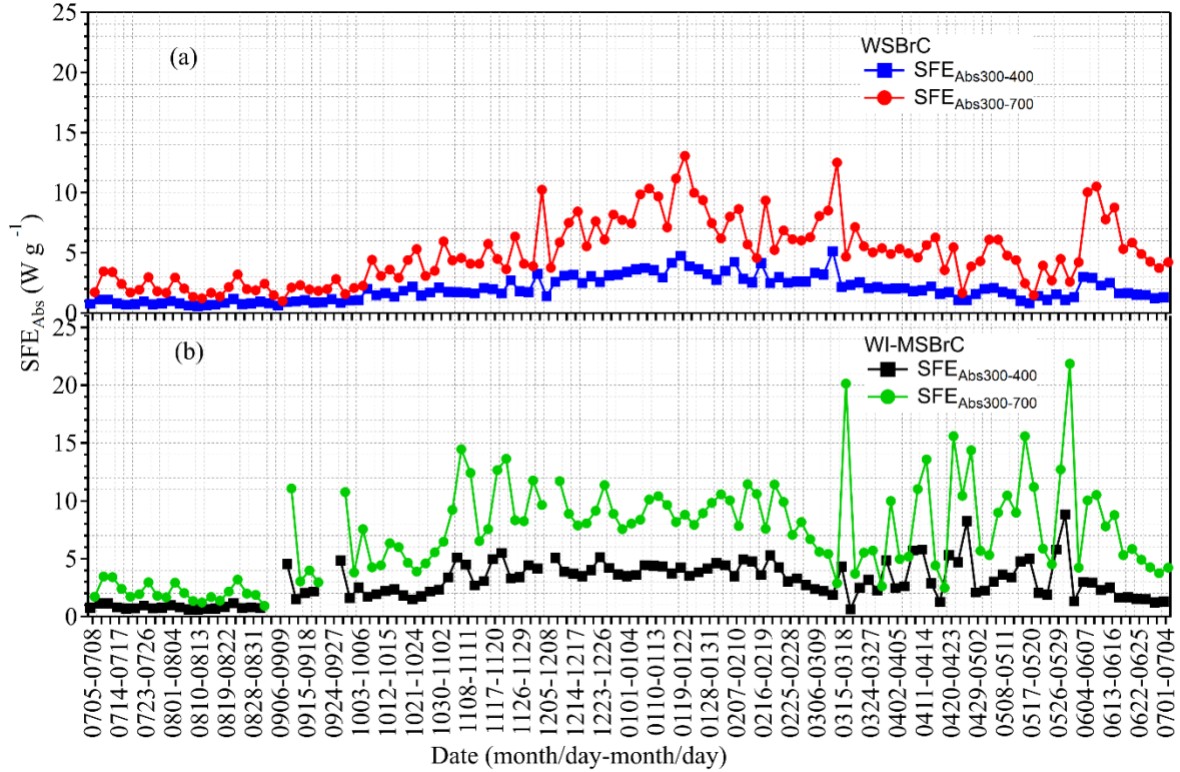


**Figure 3**. Temporal variations in SFE$_{Abs}$ of WSBrC and WI-MSBrC from 300–400nm and 300–700nm in PM$_{2.5}$ from Tianjin.

3.3 Fluorescence characteristics of WSBrC and WI-MSBrC
3.3.1 Fluorescence indices

Annual and seasonal averages of the fluorescence indices: FI, BIX and HIX, of WSBrC (FI$_{WSBrC}$, BIX$_{WSBrC}$ and HIX$_{WSBrC}$, respectively) and WI-MSBrC (FI$_{WI-MSBrC}$, BIX$_{WI-MSBrC}$ and HIX$_{WI-MSBrC}$, respectively) in PM$_{2.5}$ at Tianjin are presented in Table 1. Their ranges and median values are provided in Table S1, and temporal variations are depicted in Fig. 3. Fluorescence indices are developed as indicators for the type and source of the fluorescent organic matter (OM) in aquatic and soil systems and has been successfully applied to assess the sources and aging processes of OA in the atmosphere in recent times (Dong et al., 2023b;Lee et al., 2013;Wu et al., 2021b). FI and BIX provide the insights in exploring the source and aging of OA and received much attention in recent times (Xie et al., 2020;Gao yan and Zhang, 2018;Qin et al., 2018;Deng et al., 2022). They have been considered as indicators to assess the relative contributions of terrestrial, biological and microbially derived OM to OA. The FI values of OM lower than 1.4 indicate its terrestrial origin and the values of 1.9 or higher indicate the microbial origin, and shows an inverse relationship with aromaticity of the OM (Gao yan and Zhang, 2018;Birdwell and Engel, 2010). The BIX values of 0.8 and 1.0 correspond to the freshly derived OM of biological or microbial origin and those of ~0.6 imply the little contribution of the biological OM (Birdwell and Engel, 2010;Dong et al., 2023b). HIX reflect the degree of humification of OA, and has been considered as a proxy for aromaticity of OM and the HIX value is increased with the increasing

aromaticity and polycondensation degree (Deng et al., 2022;McKnight et al., 2001;Birdwell and Engel, 2010). The HIX values of >5 reflect the fresh OM derived from biomass and animal manure (Birdwell and Engel, 2010).

$FI_{WSBrC}$ and $FI_{WI-MSBrC}$ were ranged from 1.13 to 1.63 with an average of 1.38 and 1.29-2.24 (avg. 1.60), respectively, during the campaign in Tianjin. While $BIX_{WSBrC}$ and $BIX_{WI-MSBrC}$ were 0.79-1.39 (1.05) and 0.83-1.76 (1.26), respectively, during the campaign. Both FI and BIX of WSBrC and WI-MSBrC followed a temporal pattern, but the temporal pattern of $FI_{WSBrC}$ was exactly opposite to that of the $FI_{WI-MSBrC}$ (Fig. 4a). The $FI_{WSBrC}$ values were slightly decreased from summer to autumn followed by a gradual increase to mid-winter and then a gradual decrease to summer through spring (Fig. 4a). While the temporal variations of $BIX_{WSBrC}$ showed a gradual decrease from summer to autumn followed by a gradual increase to winter and remained relatively stable during the wintertime followed by a gradual decrease to to summer through spring (Fig. 4b). The temporal variations of $BIX_{WI-MSBrC}$ were also found to be opposite to those of the $BIX_{WSBrC}$, except in winter, in which the $BIX_{WI-MSBrC}$ values were higher compared to those in other seasons (Fig. 4b). Interestingly, the temporal patterns of $HIX_{WSBrC}$ and $HIX_{WI-MSBrC}$ were found to be similar with relatively stable in summer followed by a sharp increase in early autumn and then a gradual decrease to summer through winter and spring (Fig. 4c). Further the $HIX_{WSBrC}$ was always significantly higher than the $HIX_{WI-MSBrC}$. Such temporal differences in all the three fluorescence indices clearly indicate that the composition and/or aromaticity of WSBrC and WI-MSBrC are substantially distinct, even though they might have been mainly derived from similar sources: biomass burning and coal combustion, as discussed in previous section.

Average $FI_{WSBrC}$ was found to be higher in autumn followed the similar levels in winter and spring and the lowest in summer, whereas that of $BIX_{WSBrC}$ was higher in winter followed by autumn, spring and the lowest in summer (Table 1). While the averages of both $FI_{WI-MSBrC}$ and $BIX_{WI-MSBrC}$ were higher in winter followed by summer, spring and the lowest in autumn (Table 1). Annual and seasonal averages of FI values of both WSBrC and WI-MSBrC were around or higher than 1.4 and lower than 1.9 in Tianjin, indicating that the BrC in Tianjin was mainly derived from terrestrial OM that should have largely consist of high aromatic compounds. In contrast, the annual and seasonal averages of BIX of both WSBrC and WI-MSBrC were higher than 1.0 (Table 1), indicating the predominant contributions of OM from the biological (including biomass burning) sources. In addition, the lowest $FI_{WSBrC}$ and $BIX_{WSBrC}$ values in summer and those of the $FI_{WI-MSBrC}$ in spring and $BIX_{WI-MSBrC}$ in autumn suggest that the contribution from terrestrial sources (e.g., coal combustion) might be less in spring and autumn and the photobleaching of OA might be significant under high solar radiation in summer.

$HIX_{WSBrC}$ and $HIX_{WI-MSBrC}$ were ranged from 1.72 to 4.7 with an average of 2.87 and 0.11−2.38 (avg. 0.81), respectively, during the campaign, which again support that both the BrC components in Tianjin should have been significantly derived from biomass burning and might consist highly humified and aromatic compounds. Average $HIX_{WSBrC}$ was higher in summer followed by autumn, spring and the lowest in winter (Table 1). In contrast, the average $HIX_{WI-MSBrC}$ was higher in winter followed by autumn, spring and the lowest in summer (Table 1). It has been reported that aging processes and HIX have a significant relation (Deng et al., 2022). The higher $HIX_{WSBrC}$ and lower $HIX_{WI-MSBrC}$ in summer confirm that the BrC, which is more water-soluble, was significantly produced from aromatic compounds and subjected for significant atmospheric aging in summer over the Tianjin region.

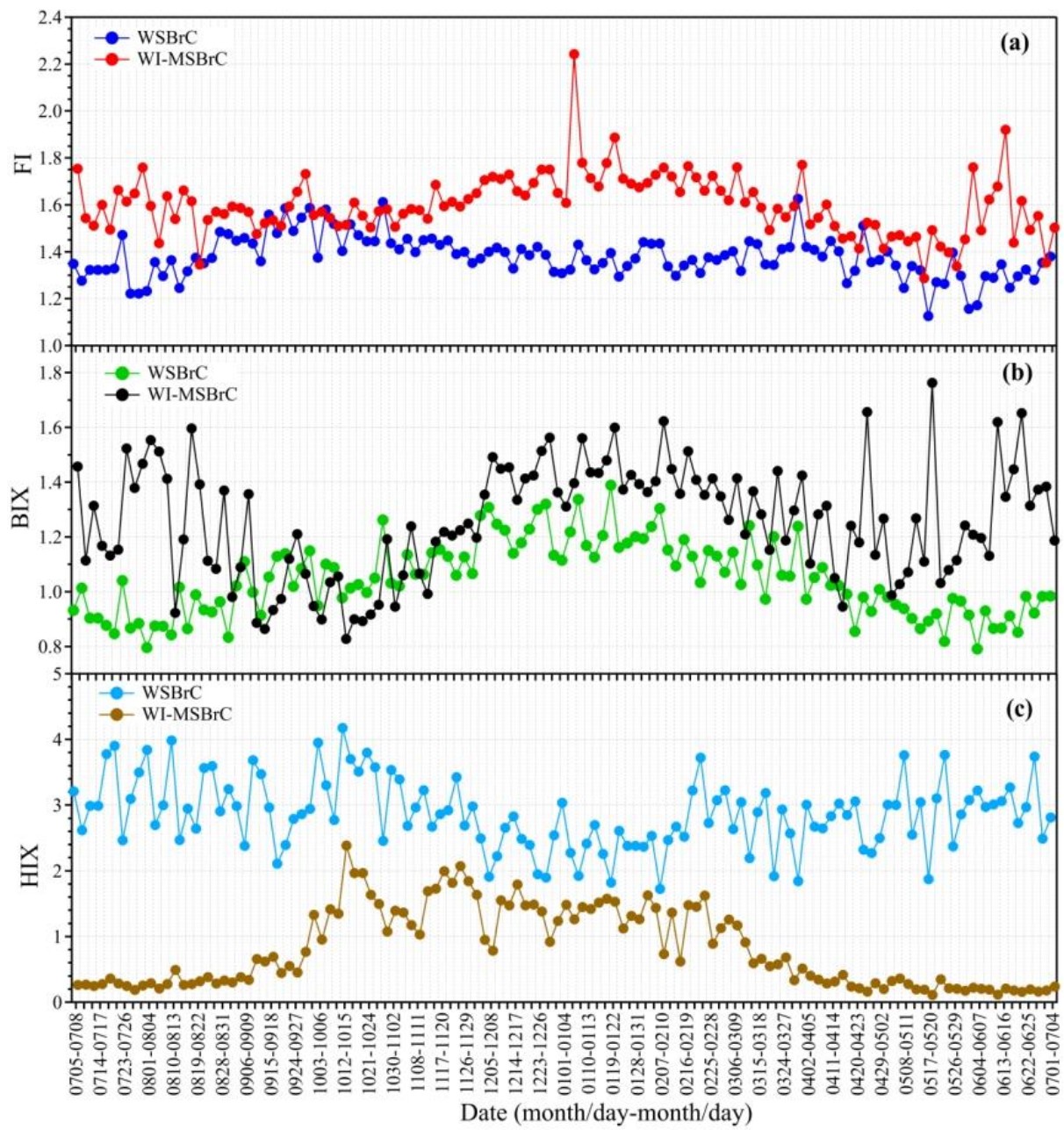

**Figure 4**. Temporal variations in light absorption and fluorescence properties of BrC in PM$_{2.5}$
Tianjin: (a) FI, (b) BIX, and (c) HIX.

3.3.2 Fluorophore identification

It is well established that fluorophores with different excitation emission wavelengths can
distinguish their types and sources. However, the types and sources of a large number of
fluorophores have not been determined due to their complex chemical composition and sources.
Here, we separated several fluorescence components from the EEM data using the parallel factor
analysis (PARAFAC) model, and the results are shown in Fig. 5. The fact of the value of core
consistency close to 100 in PARAFAC model indicates that the more the individual components

that are analyzed together make up 100% of the mixture, with no unexplained residues. The core
consistency of PARAFAC model explained the maximum variance of 89% for WSBrC and of
95% for WI-MSBrC whole data obtained during the campaign, when three independent groups
were chosen for the simulation.

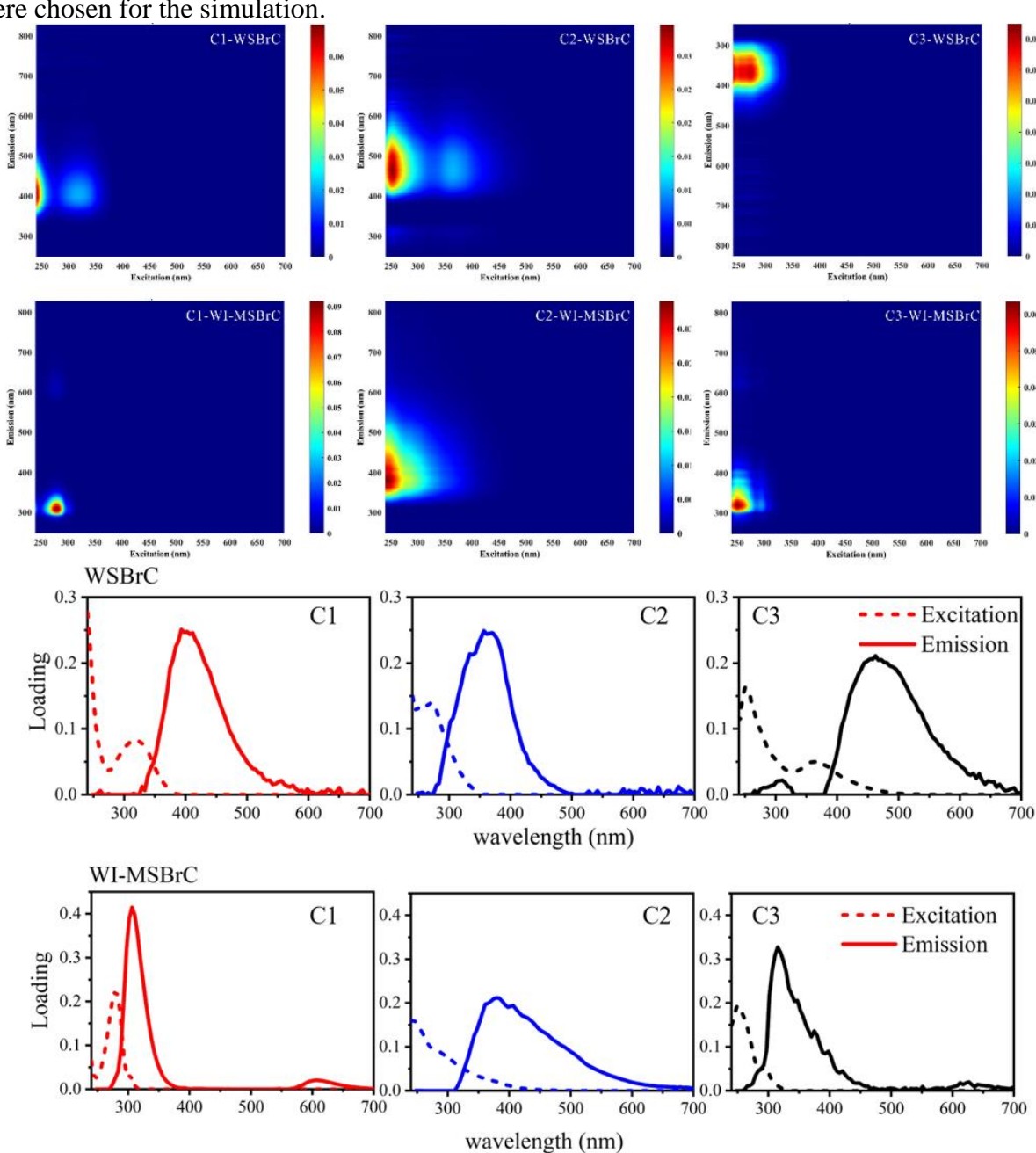

**Figure 5**. Three-dimensional excitation-emission matrix of three fluorescent components (upper
panel) together with their excitation and emission loadings (lower panel) of WSBrC and WI-
MSBrC obtained by PARAFAC model analysis.

The types of fluoroophores of both WSBrC and WI-MSBrC identified in this study together
with their excitation and emission wavelengths and those reported in the literature are summarized

in Table 2. Among the total of three types of fluorophores obtained for WSBrC in Tianjin PM$_{2.5}$ by PARAFAC for EEMs, two showed the fluorescence characteristics similar to those of less oxygenated and highly oxygenated humic-like substances (HULIS), respectively, and the third one showed similar to those of protein compounds (PLOM). Fluorophore C1$_{WSBrC}$ has a primary fluorescence peak at excitation/emission (Ex/Em): <240/393 nm, and a secondary fluorescence peak at Ex/Em: 318/393 nm. C1$_{WSBrC}$ can be classified as a humus-like fluorophore because the bimodal distribution of the fluorescence spectrum is usually associated with HULIS. The emission wavelength of C1$_{WSBrC}$ was closer to the UV region than that of the second peak of C2$_{WSBrC}$, indicating the existence of a small number of aromatic substances, conjugate systems and nonlinear ring systems (Deng et al., 2022). C2$_{WSBrC}$ (Ex/Em ~251, 363 nm/462 nm) was identified as a common HULIS in aerosols, with higher oxidation, aromatization, molecular weight, conjugation, and unsaturation due to its larger emission wavelength (Wen et al., 2021). The molecular weight of the fluorophore as well as its degree of conjugation tend to increase with the excitation wavelength, and such increase in size and the conjugation degree may be attributed to the presence of highly aromatic conjugated structures containing heteroatoms (Chen et al., 2019). Compared to C1$_{WSBrC}$ and C2$_{WSBrC}$, C3$_{WSBrC}$ also contains two peaks, with shorter wavelengths (<380 nm) emission peak, which is usually associated with protein-like organic matter (PLOM) such as tryptophan and tyrosine, with low aromatic properties and small molecular size (Table 2).

**Table 2.** Description and wavelength positions of PARAFAC components in this study and other reports from the literature. (PLOM = protein compounds; HULIS = humic-like substances)

| Category | Components | Ex(nm) | Em(nm) | Substances | References |
|---|---|---|---|---|---|
| WSBrC | C1 | <240, 318 | 393 | low-oxygenated HULIS | this study |
| | C2 | 251, 363 | 462 | high-oxygenated HULIS | |
| | C3 | <240, 271 | 356.3 | PLOM, such as tryptophan and tyrosine | |
| WI-MSBrC | C1 | <240, 279 | 306 | PLOM, tyrosine-like | |
| | C2 | <240 | 379 | uncertain | |
| | C3 | 251, 294 | 315 | PLOM, tryptophan-like | |
| Water-soluble BrC | C1 | 250, 315 | 396 | low-oxygenated HULIS | (Deng et al., 2022) |
| | C2 | 250 | 465 | highly-oxygenated HULIS | |
| | C3 | 250 | 385 | low-oxygenated HULIS | |
| | C4 | 250 | 340 | PLOM, tryptophan-like | |
| | C5 | 275 | 305 | PLOM, tyrosine-like | |
| WSOC | C1 | 240, 315 | 393 | low-oxygenated HULIS | (Wen et al., 2021) |
| | C2 | 245, 360 | 476 | highly-oxygenated HULIS | |
| | C3 | <240, 290 | 361 | PLOM, such as tryptophan and tyrosine | |
| | C4 | 275 | 311 | PLOM, tyrosine-like | |
| WSM and MSM | C1 | 255 | 415 | HULIS-1 component | (Chen et al., 2019) |
| | C2 | 220 | 340 | tryptophan-like component | |
| | C3 | 255 | 385 | HULIS-2 component | |
| | C4 | 210 | 300 | tyrosine-like component | |
| | C5 | 250 | 355 | amino acid-like component | |
| WSOC | C1 | 245 | 410 | HULIS, photodegradation of macromolecules | (Xie et al., 2020) |
| | C2 | 235 | 398 | HULIS, aromatic and saturated compounds were presented | |
| | C3 | 250, 360 | 466 | humic-like chromophores, more aromatic and consisted of more unsaturated compounds produced by condensation reactions | |
| | C4 | 250, 285 | 432 | terrestrial humic-like chromophore | |
| | C5 | <235 | 430 | terrestrial humic-like substance, photochemical product | |
| MSOC | C6 | 275 | 408 | low oxidation humic-like | |
| | C7 | 235, 275 | 372 | protein-like chromophore | |
| | C8 | 260, 310 | 364 | protein-like (tryptophan-like), may be related to PAHs | |

However, WI-MSBrC fluorophore C1$_{WI-MSBrC}$ might be tyrosine-like substance. C2$_{WI-MSBrC}$ is not quite certain and could be either HULIS or PLOM, because its emission wavelength <380 nm generally fits the profile of PLOM, but it is also close to the emission wavelength of HULIS. While C3$_{WI-MSBrC}$ is a tryptophan-like substance, which was reported to contain less aromatic and small molecular weight compounds. In general, phenols contribute significantly to C3$_{WI-MSBrC}$ fluorophore as they are the products of incomplete pyrolysis of lignin and cellulose and are used as indicators of biomass burning (Wen et al., 2021). Therefore, WI-MSBrC fluorophores of all samples in this study can be classified as mainly PLOM.

The percent contributions of each fluorophore to WSBrC and WI-MSBrC in PM$_{2.5}$ in Tianjin in each season are shown in Fig. 6. The compositions of WSBrC and WI-BrC clearly imply that the former contained more HULIS, whereas the later consist mostly of PLOM, and also indicate that most of the fluorophores of protein-like substances could dissolve in organic solvent, rather than in water.

According to the excitation emission wavelength, we classified the fluorescence component of WI-MSBrC substance as PLOM, but the correlation between their fluorescence intensity and BIX (R = 0.66, $p$ < 0.05) was very small, far lower than that of WSBrC substance and BIX (R = 0.59, $p$ < 0.05). On the contrary, the correlation between their fluorescence intensity and HIX (R = 0.74, $p$ < 0.05) was much higher than that of WSBrC (R = –0.10, $p$ < 0.05). Although PLOM may be associated with some polycyclic aromatic hydrocarbons (PAHs) or phenols from fossil fuel combustion and biomass burning, especially in urban aerosols, the correlation is puzzling.

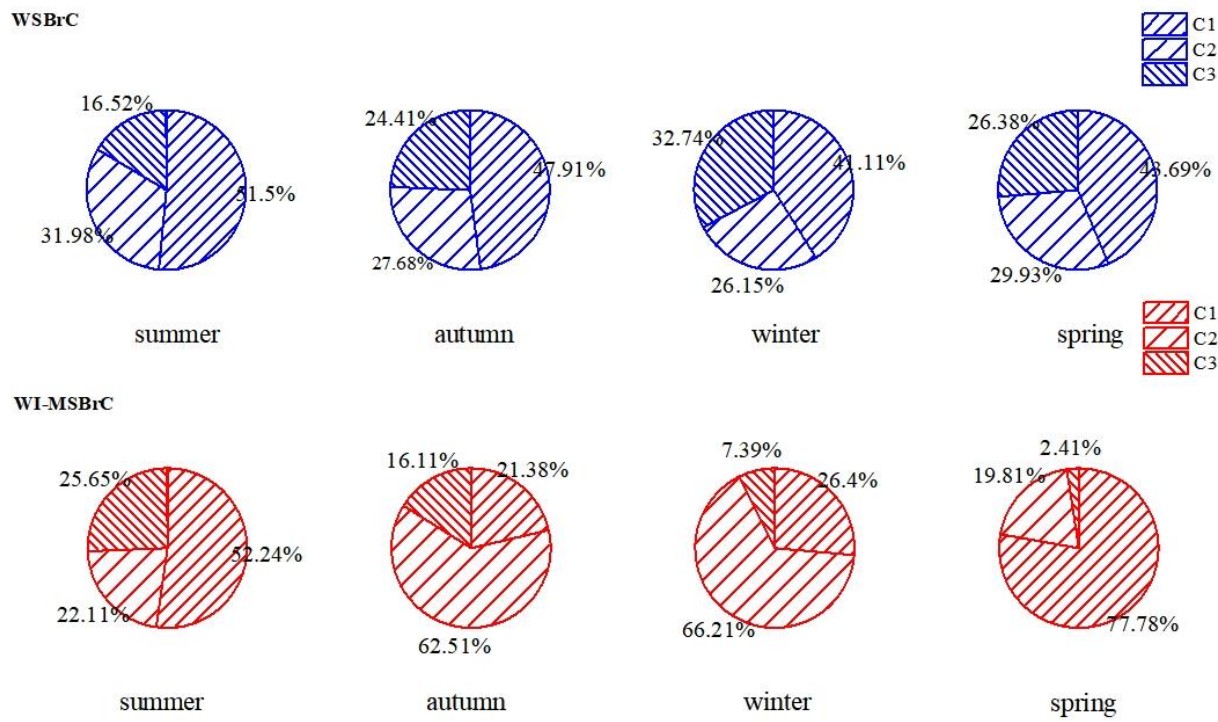

**Figure 6**. Relative abundances of the chromophores of the WSBrC and WI-MSBrC in PM$_{2.5}$ from Tianjin.

On average, the humic-like fluorophores together contributed more than 60% to the fluorescence intensity in WSBrC, suggesting that humic-like fluorophores played a dominant role

in fluorescence properties of WSBrC in Tianjin. Generally, the low-oxygenated fluorophores C1$_{WSBrC}$ made considerable contributions in each season. While C2$_{WSBrC}$, highly oxygenated HULIS, has a greater relative contribution in summer, which might be due to the strong solar radiation in summer. In contrast, in WI-MSBrC, the average contribution of PLOM to fluorescence intensity was higher than 70% in spring (80.2%) and summer (77.9%), but C2$_{WI-MSBrC}$ component dominated in winter and autumn. This indicated that biological activities increased in spring and summer and the relative abundance of bioaerosols might be higher during that period.

### 3.4 Potential sources of BrC

To further explore the potential sources of BrC, correlations of FV with chemical components and light absorption of PM$_{2.5}$ were examined. The sum of FVs of WSBrC and WI-MSBrC (FVs$_{(WSBrC+WI-MSBrC)}$) showed a significant correlation with secondary OC (SOC) in autumn (R = 0.90, $p < 0.05$) and winter (R = 0.67, $p < 0.05$). Furthermore, the correlation between FVs$_{(WSBrC+WI-MSBrC)}$ and EC in each season was insignificant. Such relations suggest that the secondary formation processes should have been played an important role in controlling the loadings of BrC in autumn and winter as well. A good correlation between FV and Abs$_{365}$ of both WSBrC and WI-MSBrC was found in all seasons except winter, which indicates that most light-absorbing materials would also have significant fluorescence characteristics.

The relative contents of different chromophores in different polar extracts depend on their sources and varied significantly. The results showed that the NFVs of WSBrC were lower than those the WI-MSBrC and were different from season to season in Tianjin (Fig. 7). Recently, it has been reported that the aerosols derived from biomass burning and coal combustion exhibit the highest NFV values, while SOA show the lowest NFV values (Chen et al., 2020). NFV in all samples studied in Tianjin during 2018–2019 was very similar to that of primary emissions and higher than that of secondary aerosols. Such result reveal that the fluorophores in the Tianjin PM$_{2.5}$ might mainly be derived from a primary combustion sources as well. In addition, the NFVs of the Tianjin PM$_{2.5}$ were higher in winter than in summer, which is likely and can be attributed to the photolysis of chromophores in summer. In addition, NFV of WI-MSBrC was much higher than that in WSBrC, which indicate that fluorescence contribution of fluorophores was abundant in WI-MSBrC than in the WSBrC.

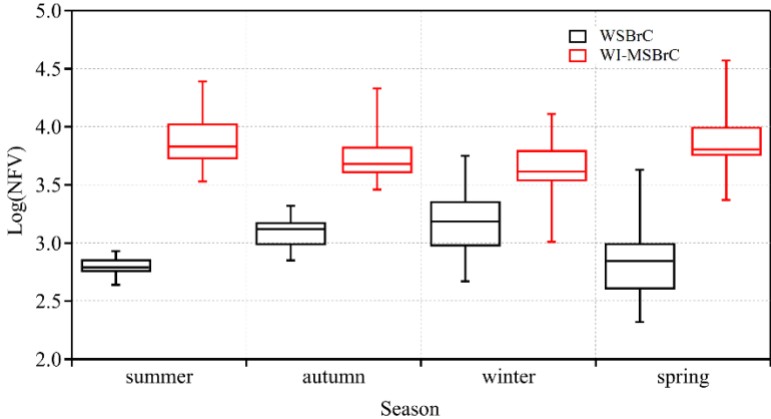

**Figure 7**. The normalized fluorescence volumes (NFVs) of the WSBrC and WI-MSBrC of PM$_{2.5}$ from Tianjin, North China.

**4. Summary and Conclusions**

This study presents the temporal variations in light absorption and fluorescence properties of water-soluble BrC (WSBrC) and the water-insoluble but MeOH-soluble BrC (WI-MSBrC) in PM$_{2.5}$ collected from Tianjin, North China during July 5, 2018 – July 5, 2019. Light absorption properties of WSBrC and WI-MSBrC in Tianjin were investigated and found to be distinct from season to season, which was lower in spring and summer, compared with that in autumn and winter. The AAE of WI-MSBrC was comparable with that of WSBrC. The mass absorption efficiency of WSBrC and WI-MSBrC (MAE$_{365}$) exhibited distinct seasonal variations, which was higher in winter and lower in summer and autumn. Biologically derived or secondary BrC and/or its photobleaching might be the reasons for the lower MAE$_{365}$ values in summer and autumn. The light absorption of both WSBrC and WI-MSBrC in the range of 300–400 nm to that in the whole range (300–700 nm) was ~40%, indicating that BrC in the UV-Vis range plays an important role in climate warming. In addition, based on PARAFAC analysis model, EEM data were comprehensively analyzed to identify the types and abundance of different fluorophores, and obtained three types of the fluorophores: low-oxygenated HULIS, high-oxygenated HULIS and protein-like compound (PLOM). The correlation between BrC optical properties and aerosol chemical composition indicated that biomass burning, and fossil fuel (mainly coal) combustion significantly contributed to BrC content in winter, while primary biological emission and subsequent aging significantly contributed to the BrC content in summer. These results illustrated the light absorption properties of BrC in metropolis aerosols and emphasized its significant contribution to radiative forcing.

**Declaration of competing intertest**

The authors declare no competing intertest in this paper.

**Data Availability Statement**

The data used in this study can be found online at https://doi.org/10.5281/zenodo.7316371 (Dong et al., 2022), and at https://doi.org/10.5281/zenodo.5140861 (Dong et al., 2021).

**Supplement.**

The supplement related to this article is available online at:

**Acknowledgments**

This work was supported in part by National Natural Science Foundation of China (Grant No. 41775120 and 42277090) & National Key Research and Development Plan (Grant No. 2017YFC0212700), China. The author also thanks to Mr. Yunting Xiao's help for writing a code to calculate the SFE.

**Author contribution**
ZD and CMP conceptualized this study. ZD and PL conducted the sampling. ZD conducted the
chemical analyses, interpreted the data and wrote the manuscript. CMP supervised the research
and acquired the funding for this study. XZ, ZXY and ZXM administrated the project. CMP, ZX,
DJ, PF and CQL contributed in discussing the results and review and editing the manuscript.

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
