# Peer review of "Measurement Report: Optical Characterization, Seasonality, and Sources of"

_EGUsphere, 2023_

## Author Comment (AC1)

Brown carbon (BrC) is a poorly characterized component of particulate matter that absorbs solar radiation and may contribute significantly to global warming. Some of the factors responsible for the current knowledge gap on BrC are the lack of understanding of its chemical composition, light-absorption properties, and contribution to total particulate mass and light absorption, which are likely subjected to significant spatial and seasonal variations. The measurement report by Dong et al. focused on the physicochemical characteristics and sources of 1BrC in Tianjin, northern China. In particular, the light absorption and excitation emission matrix fluorescence of both water-soluble and water-insoluble but methanol-soluble BrC in $PM_{2.5}$ were measured off-line using a three-dimensional fluorescence spectrometer. The measurements were performed for one year and the seasonal variations were investigated. The results showed clear seasonal differences both in the BrC light-absorption and the type of constituting chromophores, an association with the potential sources, and a significant contribution of BrC to climate warming. The methodology used was appropriated for the described investigation, and the obtained results were well presented and of high relevance to the field of atmospheric science. Therefore, I recommend for publication after the following comments are addressed:

Dear Luis Miguel Feijo Barreira,

Thank you very much for your critical reading of the manuscript, appreciation of our work and comments/suggestions, which helped to further improve the quality of the MS. The MS is revised accordingly, and our point-by-point responses to all the comments are provided below. Please see the revised MS for details of the revisions.

**Major comments: 212009**

P6L242-The authors claim that "The higher Abs365 in winter indicates that the light absorption of BrC in PM2.5 may have more significant effect on the climate and the photochemical reactions in the atmosphere over Tianjin in winter than in other seasons". However, the effect of aerosols on climate are complex and photochemical reactions depend as well on the type and amount of atmospheric oxidants, precursor VOCs, and many other factors. Therefore, these conclusions are difficult to estimate based on a higher light absorption at a specific wavelength.

*Response:* We agree with the referee's opinion. We modified this concluding point as: "Such higher abs$_{365}$, particularly in winter, indicates that BrC in $PM_{2.5}$ has a significant effect on light absorption and thus on climate system over the Tianjin region. Please see Page 7, Lines 43-46 in the revised MS.

P7254-In this sentence it is mentioned that the absorption coefficient of WI-MSBrC was always greater than that of WSBrC across the shorter wavelengths. However, In Fig. 2a the absorption coefficient of MSBrC was lower than the one of WSBrC from autumn to spring, except at a wavelength of about 280nm.

*Response:* To avoid such generalization, we modified our statement by specifying the seasons and wavelength range in the revised MS (see Page 8, Lines 10-13).

P11L301-The results show a higher correlation between Abs365 and K+ in spring and autumn, especially for WSBrC, which is opposite to what was observed for Cl- where the correlation was higher in winter. Shouldn´t the correlation with K+ be higher in winter as well when biomass burning

is usually higher? Is there any reason why biomass burning at the measured site would not be higher in winter?

*Response:* No, biomass burning is one of the major sources in winter too, but the coal combustion is a dominant source. The poor correlation of $Abs_{365}$ with $K^+$ was mainly driven by outliers in the $K^+$ concentration data, which might have been occurred due to unknown biomass burning event(s) at local scale. We noted this point in the revised MS (see Page 9, Lines 13-15).

In fact, concentration of $K^+$ was higher in winter than in other seasons, but it showed moderate correlation with $PM_{2.5}$ ($R^2$=0.47), whereas chloride ion ($Cl^-$) showed high correlation ($R^2$=0.71) with the $PM_{2.5}$ in winter, indicating that coal combustion is a dominant contributor, compared to that of biomass burning, to $PM_{2.5}$ (and for BrC as well) in winter (Dong et al., 2023).

P11L303-In this sentence, it is mentioned that the correlation between WSOC and K+ in autumn was stronger than that between MSOC, confirming that most of the chromophores generated by biomass burning were water-soluble. However, in P7L258 it says that most chromophores, including PAHs from biomass burning, were soluble in methanol. Furthermore, in Sect. 3.4 it is mentioned that "MeOH-soluble OC was much higher than that in WSBrC, which indicated that chromophores were (more?) abundant in WI-MSBrC than in the WSBrC." Can the authors clarify these differences?

*Response:* Actually, we mean that the chromophores derived from the biomass burning emissions only, but not all the chromophores, are relatively more soluble in water compared to that in organic solvent (methanol). To avoid such confusion to the reader, we made it clear modifying this sentence in the revised MS (see Page 9, Lines 15-18). Therefore, the other two statements mentioned here in the comment (P7L258 and Sect. 3.4) stand as they are.

P13L342-It is mentioned that the AAE of BrC increases with the polarity of constituents. However, the AAE of MSOC was similar to the one of WSOC. Shouldn´t the AAE be higher for WSOC?

*Response:* Generally, the AAE of BrC is increased with its increasing polarity (Chen et al. 2016). However, it has also been found that the value of $AAE_{300-600}$ of water extract of biomass burning samples is lower than that extracted into acetonitrile (Lin et al., 2017), indicating that the origin of the BrC is also play an important role. We clarified it by including this point in the revised MS (see Page 11, Lines 24-26).

P14L370-The fluorescent properties of WSBrC are dominantly described this chapter. For example, in the first paragraph of Sect. 3.3.1 the proportion of water-soluble chromophores was presented and discussed contrary to the ones of water-insoluble but methanol soluble chromophores. However, the methanol-soluble chromophores had actually contributed the most to the fluorescence volume of aerosol particles (e.g. Fig. 6). Is there a reason for focusing the discussion of this paragraph on WSBrC when the contribution of MSBrC was much more significant?

*Response:* Yes, we agree with the referee's view. We removed this paragraph and focused on both the fractions of the BrC equally throughout the text in the revised MS.

P14394- In this paragraph it is mentioned that "the higher molecular weight and aromatic organic compounds contribute more to WSBrC in summer and autumn while the contents of WI-MSBrC (winter > autumn > spring > summer) were opposite". Can the authors clarify in more detail the results that supported this conclusion?

*Response:* We substantially improved the discussion about all the three fluorescence indices in the revised MS (see Section 3.3.1).

**Minor comments:**

P1L33-The word "However" should be replaced by "For example," since that sentence does not contradict the previous one.

*Response:* We replaced the word "However" with "For example" in the revised MS (see Page 1, Line 34).

P2L41-Do the authors mean nitrogen-containing aromatic compounds?

*Response:* We mean that the compounds with polar functional group that consists of nitrogen and/or oxygen. We modified the phrase to make it clear to the reader in the revised MS (see Page 2, Lines 1-3).

P2L48-Automobile exhaust is a consequence of fossil fuel combustion. Could these be combined?

*Response:* We removed the phrase 'automobile exhaust' in the revised MS.

P2L55- Not only ultraviolet spectroscopy but ultraviolet-visible spectroscopy is commonly used to study the light absorption of brown carbon.

*Response:* We changed "ultraviolet spectroscopy" into "ultraviolet-visible spectroscopy" in the revised MS (see Page 2, Line 27).

P2L67-Can the authors clarify this sentence? In particular, are the authors comparing the sensitivity of EEM with the one from UV-vis spectroscopy? What type of classification are the authors referring to and what "shape of absorption spectra" means in this case?

*Response:* No, we are only highlighting the advantage of the fluoresce method by referring the absorption spectral measurements. We modified the corresponding phrases to clarify the type (chromophores) of classification and to avoid the confusion about the shape of spectra in the revised MS (see Page 2, Lines 41-43)

P2L73-This sentence could be changed to "quantitative measurement of light-absorbing organic components". The structural properties of those compounds are also important since they determine both the light-absorption and the potential health effects.

*Response:* We removed this part of text in the revised MS.

P2L74- There are currently ways to differentiate between light absorption of black and brown carbon. For example, this differentiation has been done using AE33 measurements and employing a method based on the wavelength dependence of AAE (WDA analysis). However, it is true that this separation is challenging since both components absorb light over the entire UV-vis range and some assumptions are made to separate their light absorptions.

*Response:* Yes, we agree with the referee. However, we removed that part of text in the revised MS.

P4L133-Can the extraction method be described in more detail? For example, was the entire filter used for the extraction or a part of it (1.0 cm by 1.5 cm in size?) was taken for the OC/EC analysis? Were the filters weighted prior and after particle sampling? Where were the quartz filters inserted

during ultrasonic extraction? The PTFE filter was used to remove undissolved particles, right? Currently, it is mentioned that it was used to remove water-insoluble compounds. The majority of those should remain in the extraction vessel/tube for subsequent extraction with methanol. Was the extraction efficiency determined in this study?

*Response:* Following the referee's suggestion, we improved the method description substantially by including all these details and provided the citations for further details in the revised MS (see Sects. 2.1 and 2.2). Yes, the extracts were filtered through PTFE syringe filter to remove the undissolved particles. We corrected it in the revised MS (see Page 4, Line 20). Since it is a well-established procedure, we didn't check the extraction efficiency in this study.

P4L145-Should this section be part of the chemical analysis? I understand that the used instrument relies on the measurement of a physical property, but it is an analytical chemistry instrument. Or Sect. 2.2. could be changed e.g. to "Physical-chemical analysis".

*Response:* We agree with the referee's opinion. However, in order to keep the main attention on optical properties, we rephrased the sub-titles and also re-structured the chemical analysis part in the revised MS (see Sects. 2.2 and 2.3).

P4L154-These samples were also analysed by the HORIBA Aqualog optical spectrometer, right? In that case, there is no need to mention "a fluorescence spectrometer" since the blank samples were analysed by the same instrument.

*Response:* Yes. We removed the phrase: "a fluorescence spectrometer", in the revised MS.

P4L170-The AAE can both indicate that the BrC has a greater of smaller contribution to aerosol absorption, depending on its value. The last part of the sentence can be removed.

*Response:* As suggested, we removed the last part of the sentence in the revised MS.

P6L219-The units can be removed from the title.

*Response:* We removed the units form the title in the revised MS.

P6L227-The decomposition of BrC constituents during summer, likely due to photobleaching, is induced both by solar intensity and oxidants present in the atmosphere.

*Response:* We modified this sentence accordingly in the revised MS (see Page 7, Lines 14-16).

P6L229-This is true, but a decrease in atmospheric oxidation likely plays a role as well.

*Response:* We included this point in the revised MS.

P7L261-The nitroaromatics compounds do not always contribute 60% to the absorbance. The sentence needs to be rewritten to e.g. "In another study, nitroaromatics have contributed 60 % to the total absorbance in the 300-400 nm range".

*Response:* We modified it accordingly in the revised MS (see Page 8, Lines 17-19).

P8L282-The authors mean on summer holidays? This paragraph can be combined with the previous one because it continues that discussion.

*Response:* Yes. We replaced that phrase with summer to avoid any confusion to the reader and combined the two paragraphs in the revised MS (see Page 9, Lines 3-6).

P8L283-And photobleaching as mentioned earlier?

*Response:* We included the word, 'Photobleaching' in the revised MS (see Page 9, Line 4).

P12L316- The references are missing.

*Response:* We cited appropriate references in the revised MS (see Page 11, Lines 2-3).

P13L320-Can the authors clarify this sentence?

*Response:* We mean that unlike the direct measurement of AAE of the particulate matter, which is influenced by factors such as particle size, mixing state and chemical composition, the AAE of the solution depends on only the chemical composition. We made it clear in the revised MS (see Page 11, Lines 10-15).

P13L343-The large MAE365 during winter is a consequence of air pollution. Therefore, the sentence should be modified to "which is a result of severe air pollution in the mentioned period".

*Response:* We modified this section substantially in the revised MS.

P13L45-This sentence can be deleted or moved to the Methods section.

*Response:* We moved this sentence to the Methods section in the revised MS.

P13L347-Change to UV-Vis range.

*Response:* Modified it in the revised MS (see Page 12, Line 40).

P13L350-What were the values for MSBrC?

*Response:* We included the values of WI-MSBrC in the revised MS (see Page 13, Lines 18-20).

P13L355-Are the presented values the SFE range?

*Response:* Yes, the values are the SFE range.

P13L359-Change to UV-Vis range.

*Response:* We changed it to "UV-Vis range" in the revised MS.

P21L514-Can the authors clarify this paragraph? Is this the total FV of SOC (WSBrC+WIBrC) or the correlation of FVs of WSBrC and WIBrC with SOA?

*Response:* What we meant the total FVs was that the sum of FV of WSBrC and WI-MSBrC. To avoid such confusion to the reader, we modified this phrase in the revised MS (see Page 20, Lines 8-10).

P21L520-The second sentence can be removed/modified since Fig. 10 refer to the NFVs and not to the overall optical properties of the different samples.

*Response:* We agree with the referee and modified the first and second sentences of this paragraph in the revised MS (see Page 20, Lines 16-18).

P22L540- N-containing substances were soluble in water. but how about in methanol?

*Response:* Since stable carbon isotope ratios did not show any relations, we removed this part of discussion to avoid any mislead from the drawn conclusions in the revised MS.

P2L5553-Do the authors mean that Tianjin PM2.5 contained more polar BrC than other cities of

China as shown by the higher AAE values?

*Response:* We removed this phrase in the revise MS.

P23L556-And photobleaching?

*Response:* We added this phrase in the revised MS (see Page 21, Line 10).

P23L568.The polarity of water is higher than the one of methanol. Since WIBrC contributed the most to the fluorescence of aerosols, can the authors conclude that this indicates that there were more polar BrC substances in the collected aerosol samples?

*Response:* We removed this sentence in the revised MS.

p23L571-How about biomass burning?

*Response:* Yes, biomass burning and coal combustion emissions are the major sources in winter. We corrected it in the revised MS (see Page 21, Line 17).

---

## Author Comment (AC2)

**Review of** "Measurement Report: Optical Characterization, Seasonality, and Sources of 2 Brown Carbon in Fine Aerosols from Tianjin, North China: Year-round Observations" by Dong et al.

**Review prepared by**: Dr. Taveen Kapoor and Dr. Rajan Chakrabarty

This study reports the optical properties of water-soluble and methanol-soluble brown carbon (BrC) in fine aerosols ($PM_{2.5}$) sampled over a year in Tianjin, North China. The authors employ a relatively new BrC characterization technique, three-dimensional fluorescence spectroscopy measure the seasonal variations in optical properties of BrC and their chromophore constituents. They evaluated the relationship between BrC and chemical composition in $PM_{2.5}$ and the possible sources of BrC over the sampling region. Overall, the manuscript needs revisions before it can be considered for publication as a measurement report in ACP.

Dear Dr. Taveen Kapoor and Dr. Rajan Chakrabarty,

Thank you very much for your critical reading of the manuscript, appreciation of our work and comments/suggestions, which helped to further improve the quality of the MS. The MS is revised accordingly, and our point-by-point responses to all the comments are provided below. Please see the revised MS for details of the revisions.

**Major comments:**

1) The novelty of this study is in the use of relatively new characterization techniques of BrC. Because of the relative newness, there is a need for justification and more context as to why the authors chose this technique over previously established ones.

The excitation-emission matrix technique is used to understand the fluorescence spectra of organic carbon compounds. The reason for studying the fluorescence of organic compounds is not clear. Also, a range of expected values for the different chromophore groups before indicating the measured values will help the reader to interpret the results better. For example, (L405) a fluorescence index < 1.4 indicates higher aromaticity but later it is said that BrC contain mainly aromatic compounds (L415) despite values of up to 2.23 (L412).

The PARAFAC analysis is used to identify chromophore groups. Three groups have been identified, but information about why there are three groups and not more (or less) has not been mentioned. The selection procedure should be mentioned.

Information on how the humidification index, δ15NTN and δ15CTC are calculated is missing from the manuscript.

*Response:* Following the referees' comment, we provided the justification/purpose of the use of three-dimensional fluorescence spectroscopy and extraction of the water-soluble and water-insoluble fractions of BrC in the introduction section in the revised MS (see Page 2, Line 14 to Page 3, Line 10).

The reason for the use of EEM technique in this study is to identify the fluorophores and thus the molecular composition of BrC. This point has been clarified in the revised MS introduction section (see Page 2, Lines 27-44). Also provided the reference values of FI, BIX and HIX in Section 3.3.1

of the revised MS.

We clarified the procedure followed to choose 3 fluorophores in PARAFAC analyses in the revised MS (see Section 3.3.2).

We have added the calculation method of the humidification index in the revised MS(see Section 2.3). Since the correlations with isotope data are not significant, we removed this part in the revised MS to avoid any ambiguity.

2) While considerable effort has been put into generating and summarizing the data, the study fails to connect the measurements to provide a coherent picture from the results of the various measured properties (BrC concentrations, absorption, fluorescence, fluorescence, humidification, etc.). Some attempts are made to correlate the two sets of properties using linear correlations, but these do not necessarily lead to consistent results. For example, L302 says that biomass burning is a major source of BrC in the autumn months, but L409 says that terrestrial organic matter is the major source. These apparent contradictions make the manuscript difficult to understand, which may be avoided by providing appropriate context to the measurements being made, as highlighted in the previous comment. A discussion of the interrelationships between the measured properties, with a special emphasis on the new findings from the new measurements will be of great benefit to the scientific community.

*Response:* Following the reviewer's suggestion in this and previous comment, we revised the MS with substantial improvement in both introduction and results and discussion. Also, we made it clear that the BrC in Tianjin is mainly contributed from mixed (biomass burning and coal combustion) sources based on the fluorescence indices data (see section 3.3.1).

3) L400 and L415 make strong statements about aging of BrC based on the measured humidification index and fluorescence index. The authors are requested to substantiate their claims, as arguments do not seem convincing in their present form, i.e., without any direct measurements of aging.

*Response:* Following the reviewers' suggestion, we completely revised this section and discussed the possible sources (by tone downing about aging) based on the humidification index (BIX) and fluorescence index (FI) results and their reference values available in the literature.

4) A lot of information provided in the figures and tables may be moved to the supplementary material as the information they provide and the text discussing them is disproportionate. Figures 3, 4, and 11 show scatter plots amongst various measured properties. But most of the discussion on these figures is around just the correlation coefficients and not the actual values (which are summarized in Table 1 already). The authors may consider moving some of the figures to the supplementary material or use a more concise figure to summarize the same information. Similarly, Figures 1, 5, and 7 show time series but the discussion is restricted to seasonal variations.

Table 1 reports the mean and standard deviations of the measured values and the range of values. Providing both sets of information seems redundant, and one set may be moved to supplementary material. Some of the information in these tables is also repeated in Figure 2.

Figure 8 has two kinds of plots, but the set of line plots is not labelled, and it is unclear what they represent. These likely show the emission and excitation spectra, which are already shown in the

figure set of three-dimensional figures above.

*Response:* We agree with the reviewers' opinion. Figures 3, 4, 6 and 11 of previous version of the MS are moved into the supplement of the revised MS.

As suggested, only mean and standard deviation data is kept in Table 1 and the ranges and median are provided in the supplement to make the full summary available for the reader in the revised MS. The annual summary and temporal variations are described and discussed, in addition to the seasonal variation in the revised MS. To avoid repetition of the data, Figure 2b-d panels are removed in the revised MS.

Yes, the linear plot in Figure 8 (Fig. 5 in the revised MS) shows the excitation emission wavelengths of the different fluorophore groups. We added annotations and kept in the revised MS to make the volumes more clear to the reader.

5) Differences are reported between the parameters measured during the different seasons, but the statistical significance of the differences are not discussed. These should be added to make the discussion more robust.

Since the authors are checking for associations between the variables, the R value should be reported instead of the R2 value (used as a measure for model predictability). Also, the significance of the correlations reported in L487-489 do not seem correct (R2= 0.01/0.06, having significant correlations, p < 0.05). These need to be re-checked.

*Response:* We improved the discussion about fluorescence indices and other parameters significantly but limited the discussion on correlations between them, because the obtained correlations are not strong (just weak to moderate). We re-checked the *p*-value and found that they are statistically significant (p <0.05), despite weak to moderate correlations, probably due to large dataset.

**Minor comments:**

- Abstract could include context on the need for the measurements.

  *Response:* We added the context of need of this study in abstract of the revised MS (see Page 1, Lines 10-12).

- L120: "The blank filters were left in the filter hood for 10 minutes" - not clear what this means and why this was done.

  *Response:* We collected the filter blanks to correct the results from the procedural errors / contamination, by placing the filter in hood for 10 mins without turning on the pump. We made it clear to the reader in the revised MS (see Page 3, Lines 42-44).

- L125: OC and EC are not spelled out before first use here.

  *Response:* We added the full form of OC and EC in the revised MS (see Page 4, Line 5).

- L120: Should be "thermal-optical carbon analyzer".

  **Response:** We corrected it in the revised MS (see Page 4, Line 8).

- L144: How was the concentration of WSOC determined? In the equation it is implied that all the organic compounds that are water insoluble OC are methanol soluble. While this

may be a fair assumption, the authors may acknowledge that there may be some chromophores that are also methanol insoluble (Shetty et al., 2019).

*Response*: The concentration of WSOC was determined using a total organic carbon analyzer, with the specific experimental method described in our previous paper (Dong et al., 2023;Wang et al., 2019). We noted this in the revised MS (see Page 4, Lines 5-11).

We agree with the reviewers' view that all water-insoluble organic compounds are not soluble in methanol. We noted this point in the revised MS to make our assumption fair (see Page 4, Lines 29-31).

- L159: Please provide a reference for this equation.

  *Response:* We cited the references in the revised MS (see Page 5, Line 4).

- L170: Line is unclear and needs to be rephrased.

  *Response:* We modified this expression in the MS (see Page 5, Lines 15-17).

- L174: 'C' is a constant and not the concentration of extract"

  *Response:* We regret for this mistake and corrected it as "a composition-dependent constant" in the revised MS (see Page 5, Line 19).

- L185: Please provide a reference for the equation.

  *Response:* As suggested by the reviewer, we cited references for the equation in the revised MS (see Page 5, Line 30).

- L202: PARAFAC and SOLO are not defined before first use here. Please also provide a link or reference to the code here.

  *Response:* We defined them in the introduction in the revised MS (see Page 2, Lines 27 and 33-34). Also, noted the SOLO model in the revised MS (see Page 6, Line 4).

- L214: What is basis of selecting the values of constants in the in the equation to calculate the SFE? Also, a value for backscatter coefficients is provided but is not present in the equation. Why is the mass scattering efficiency being ignored?

  *Response:* We chose the values of constants in the equation according to the previous literature (Chen and Bond, 2010;Deng et al., 2022;Tian et al., 2023), and the scattering efficiency is ignored, considering that BrC impacts the radiative effects by light absorption only. We noted this point in the revised MS (see Page 6, Lines 24-26).

- L230: Comparison of measured absorption those reported at sites in the USA, but these are likely to be influenced by very different source. There are several other studies reporting BrC absorption in areas with biomass and coal combustion sources that may offer a fairer comparison.

  *Response:* We agree with the reviewers' view. We compared the data from USA as well in order to assess potential influence from fossil fuel combustion emissions, in addition to biomass burning and coal combustion. We have improved the discussion substantially by including the absorption coefficient (Abs) of brown carbon from different sources in the revised MS (see Page 7, Lines 12-46).

- L254: "...absorption coefficient of WI-MSBrC was always greater than that of WSBrC across the shorter 256 wavelength..." - this is not true for spring, summer, and autumn months!

  *Response:* We corrected it by specifying the particular seasons and wave length range in the revised MS (see Page 8, Lines 10-14).

- L303: Is there a known source of biomass burning during the autumn season that can corroborate this result?

  *Response:* No, there is no specific known source of biomass burning in these seasons. However, our other study on $\delta^{13}C_{TC}$ and $\delta^{15}N_{TN}$, as well as the seasonal variation of $K^+$ and $Cl^-$ concentrations indicated that biomass burning and fossil fuel (coal) combustion are the major sources of carbonaceous aerosols in autumn and winter. We cited this reference in the revised MS (see Page 9, Lines 10-13).

- L308: not clear what is meant by "dust in spring"

  *Response:* To make the discussion clear here, we modified it and removed that (dust) phrase in the revised MS (see Page 9, Lines 18-22).

- L330: The finding that AAE_WSOC and AAE_WI-MSOC are similar is a bit surprising since the water insoluble portion is expected to have a stronger absorption and weaker wavelength dependence (please see Saleh, 2020, and references therein).

  *Response:* Yes, Saleh et al. (2020) has reported four categories of brown carbon and their extraction efficiency in water and organic solvents and the differences in AAE (Saleh, 2020). It has also been reported that the AAE values of the water extract are greater than those of the acetone and methanol extracts (Shetty et al., 2019), and interpreted that the extraction efficiency of polycyclic aromatic hydrocarbons from methanol or other organic solvents is higher than that from water, leading to a higher absorption at longer wavelengths in the methanol extract and therefore a lower AAE value.

  However, our extraction procedure is different (first with water and then with MeOH) from that of Saleh et al. (2020). We re-calculated and re-checked the data and found that the reported data here is correct. In fact, Li et al. (2020) also reported the mean value of AAE for BrC dissolved in acetonitrile in Xi'an is 6.04, while that for water-soluble brown carbon is 5.11 in winter (Li et al., 2020).

- L358: The results demonstrate the radiative forcing from BrC absorption. Whether or not they contribute significantly to radiative forcing depends on the overall radiative forcing magnitude – please rephrase. The same also needs to be edited in the abstract and summary sections.

  *Response:* We modified the expression by changing it to "UV-Vis range" in the revised MS (see Page 13, Lines 22-25).

- L373: It is not clear how a comparison between water soluble and water insoluble would lead to seasonal differences in the remaining part of the sentence.

  *Response:* No, it was a language error. In fact, we removed that text, focusing on the discussion of fluorescence indices in this section of the revised MS.

- L376: "more water-soluble chromophores" - relative to what?

  *Response:* We removed this text in the revised MS.

- L382: Was SOA calculated or was this a finding from a previous study? Please provide details or a reference.

  *Response:* We removed this text in the revised MS.

- L392: Seems to be a typo here – please rephrase.

  *Response:* We fully modified this section and took care for any language errors in the revised MS.

- L416: What are terrestrial organics?

  *Response:* We used this phrase, terrestrial instead of anthropogenic, based on the terminology given in the literature (Birdwell and Engel, 2010), which refer here is "not biological and microbial" derived organics. We mentioned it including the reference values of indices in the revised MS (see Page 14, Lines 13-21).

- L443-445: What is meant by core consistency and unexplained residues?

  *Response:* When SOLO data processing software performs PARAFAC analysis, it needs to evaluate the accuracy of model analysis results through core consistency, which is a parameter configured by the software itself. When we input the number of types of fluorophores, the higher the core consistency, the higher the accuracy of model analysis, and no other residues are not resolved. In other words, it is the percent of variance explained in the data set.

  The unexplained residue refers to the missing fluorescent chromophores. When the core consistency reaches 100%, the resolved chromophores together constitute 100% of the mixture, and no other compounds are left out.

- L455: PLOM not defined before its first use.

  *Response:* We defined it in the revised MS (see Page 18, line 16).

- L471, 475: Please provide a reference.

  *Response:* We determined the chromophore types according to the excitation and emission wavelengths summarized in Table 2, and the relevant references have been listed in Table 2. We cited the Table 2 here in the revised MS (see Pages 18 & 19, Lines 17 & 2, respectively).

- L278: SOC not defined before its first use.

  **Response:** We defined it in the revised MS (see Page 20, Line 9).

- L554: It is unclear how a similar AAE of WS and WI BrC implies that that the more polar.

  *Response:* We removed that phrase in the revised MS.

- L563: What is meant by color clusters?

  *Response:* We corrected it to fluorophores in the revised MS (see Page 21, Line 14).

- L567: reference?

*Response:* Since it is not a concluding point here, we removed it in the revised MS.

---

## Author Response (AR2)

Dear Dr. James Allan,

We thank you and referees very much for critical reading of the manuscript and comments / suggestions, which helped to improve the quality of the MS further. The MS is revised according to all the comments, and our point-by-point responses are provided below. Please see the revised MS for details of the revisions. The page numbers noted here correspond to the revised MS with track changes version.

**Editor**

Please see the further comments by the reviewers. Much of this is largely technical and should be fixed before publication. In the specific case of SFE raised by reviewer #1, it is certainly true that the scattering effect of BrC will be very important in the atmosphere when considering the instantaneous radiative forcing, however I also recognise that this will be impossible to constrain using the sampling methods described here because information on the specific size of the particles will be lost, on which the scattering has a stronger dependency than the absorption. Instead, I would invite the authors to make clear in this work that the SFE presented is the absorption component only (maybe identifying with a subscript) and explaining that a complete evaluation of the SFE would also contain an evaluation of the scattering, which is likely to be substantial.

*Response:* We completely agree with you. Following your suggestion, we clarified that the reported SFE is only the absorption component and presented it as "SFE$_{Abs}$" in the revised MS (please see Section 2.3.3 (Lines 267-270), Table 1, Figure 3 and Section 3.2).

**Referee #1**

The authors have addressed most of the questions and concerns raised during the first round of review. However, there is one confusing but important point regarding their calculation of specific forcing efficiency SFE), which only considers absorption from BrC and ignores scattering. It would help to show how absorption from BrC changes with respect to a non-absorbing OC, but that's not how it is presented in the manuscript. Therefore, the results can be misleading.

*Response:* We thank the referee for his/her critical reading of the manuscript, appreciation of our work and constructive comments/suggestions.

To avoid any confusion or misleading the reader, we clarified that the SFE presented in the revised MS correspond to only absorption component (please see Section 2.3.3 (Lines 267-269), Table 1, Figure 3 and Section 3.2).

**Specific Comments:**

- In the specific forcing efficiency calculations (Section 2.3.3), the values for constants are chosen based on previous studies (as written in the response document), but these are not referenced in the manuscript.

*Response:* We calculated the SFE using the constants reported by Deng et al (2022). We cited this reference at appropriate place in the revised MS (see Line 265).

- Also, in the specific forcing efficiency calculations it is stated that brown carbon particles only affect direct radiative forcing through absorption and ignore the scattering component. This is not a fair assumption as these particles typically have significant single-scatter albedos (of at least 0.4), meaning their scattering coefficients cannot be ignored. Ignoring the scattering would overestimate the SFE from BrC.

*Response:* We fully agree with the referee. To avoid any such error / misreading, we clarified that the reported SFE is only the absorption component and presented it as "SFE$_{Abs}$" in the revised MS (please see Section 2.3.3 (Lines 267-270), Table 1, Figure 3 and Section 3.2).

- P7L30: Absorption units are missing.

*Response:* We added the units in the revised MS (see Line 316).

- The authors could consider adding the short discussion regarding the AAEs of WS and WI-MS OC being similar (from the response document), to the manuscript.

*Response:* Following the referee's suggestion, we added discussion regarding the AAE of WSBrC and WI-MSBrC in the revised MS (see the lines 417-425).

- P11L27: typo in this sentence "...from 2.08 12.9 (ave. 6.06)..."

*Response:* We corrected it in the revised MS (see the line 414), and throughout the text.

**Referee #2**

The paper is interesting and I find it useful. The authors have replied sufficiently to the questions of the reviewers. However, I still have some minor revision wishes of my own.

*Response:* We thank the referee for his/her critical reading of the manuscript, appreciation of our work and constructive comments / suggestions.

1) You have several equations, just written in the text. Use equation editor, give them proper equation numbers and refer to them in the text.

*Response:* Following the referee's suggestion, we used the equation editor for equations and numbered them in the revised MS (please see Section 2.3).

2) p.5, lines 24-32. You present MAE and MAC. But the are the same thing, some authors simply prefer using MAC, some MAE. Make up your mind and be consistent. Reading the paper I see you have mainly used MAE so it would make sense to change the few occasions of MAC to MAE.

*Response:* Following the referee's advice, we replaced MAC with MAE, unifying the abbreviation of mass absorption efficiency, throughout the text in the revised MS.

3) p. 6, lines 8-14. Present also FI, BIX and HIX as equations with eq. numbers. Give references to all of them, this is not the first paper on them.

*Response:* Following your suggestion, we added the references for FI, BIX and HIX and presented them as equations in the revised MS (see Section 2.3.2).

4) p. 12, line 20-21. You present E2/E3. Give also that as an equation with a number.

*Response:* We modified it into the equation form in the revised MS (see eq. 5 in Section 2.3.2).

5) There is no uncertainty analysis, not even the word "uncertainty" in the paper. Use error propagation to calculate uncertainties of the different quantities that you have calculated.

*Response:* Since we measured the light absorption and emission and then calculated all the optical parameters following the standard procedures reported in the literature, and not used any authentic standards, we did not consider estimating the uncertainty through error propagation. However, we included the uncertainty found in the measurements of carbonaceous and ionic components in the revised MS (see Section 2.2).

---

## Author Response (AR3)

**Response Letter**

**Comment**

Thank you for making the modifications as requested by myself and the reviewers. Before publication, I would like to provide the authors the opportunity to improve the modified text in the third paragraph of section 2.3.3 (starting on line 267 in the 'tracked changes' version). The language is currently difficult to read and may be construed as misleading. It should be stressed that BrC can effect a strong instantaneous negative forcing through scattering, however it is not possible to evaluate this directly using offline samples because of the strong dependency on particle size.

*Response*

Dear Dr. James Allan,

We thank you very much for accepting our paper for publication in ACP and providing an opportunity to modify a part of text in Section 2.3.3 in the manuscript.

We modified it as: "*It is important to note that BrC can affect a strong instantaneous negative forcing through scattering, however it is not possible to evaluate this directly using offline samples because of strong dependency on particle size. That is why, we limited to estimate the radiative effect caused by only the absorption component of the BrC in this study.*", in the final version of the MS (please see Page 6, Lines 251-254).

We also made minute text modifications in Figure 5 caption and added "North China" in Figure 6 caption in the MS and corrected minor errors/typos in Figure S1 & S2 captions in the supplement.